# Mechanistic within-host mathematical model of inhalational anthrax

**Bevelynn Whaler**[ID][1]*, **Grant Lythe**[1], **Joseph J. Gillard**[2], **Thomas R. Laws**[ID][2], **Jonathan Carruthers**[3], **Thomas Finnie**[ID][3], **Carmen Molina-París**[1,4], **Martín López-García**[1]*

**1** Department of Applied Mathematics, School of Mathematics, University of Leeds, Leeds, United Kingdom, **2** CBR Division, Defence Science and Technology Laboratory, Salisbury, United Kingdom, **3** Data, Analytics and Surveillance, UK Health Security Agency, Salisbury, United Kingdom, **4** T-6, Theoretical Biology and Biophysics, Theoretical Division, Los Alamos National Laboratory, Los Alamos, New Mexico, United States of America

* B.Williams1@leeds.ac.uk (BW); M.LopezGarcia@leeds.ac.uk (ML-G)

**Data availability statement:** Computer codes (in Python) for reproducing the results in Figs 2–6 and 8–13 are available from the GitHub repository: https://github.com/Bevelynn/within-host-anthrax-model.

## Abstract

We present a mathematical model of the dynamics of *Bacillus anthracis* bacteria within the lymph nodes and blood of a host, following inhalation of an initial dose of spores. We also incorporate the dynamics of protective antigen, which is the binding component of the anthrax toxin produced by the bacteria. The model offers a mechanistic description of the early infection dynamics of inhalational anthrax, while its stochastic nature allows us to study the probabilities of different outcomes (for example, how likely it is that the infection will be cleared for a given inhaled dose of spores) in order to explain dose-response data for inhalational anthrax. The model is calibrated via a Bayesian approach, using *in vivo* data from New Zealand white rabbit and guinea pig infection studies, enabling within-host parameters to be estimated. We also leverage incubation-period data from the Sverdlovsk 1979 anthrax outbreak to show that the model can accurately describe human time-to-symptoms data under reasonable parameter regimes. Finally, we derive a simple approximate formula for the probability of symptom onset before time $t$, assuming that the number of inhaled spores has a Poisson distribution.

## Author summary

We use a stochastic model to characterise *Bacillus anthracis* infection in rabbits and guinea pigs. The model quantifies the kinetics of pathogenesis in different compartments, which represent different areas that the infection spreads to. It allows us to account for the large variability in infection timecourses between individual animals exposed to a high aerosol dose of spores, and to capture the probability of infection following exposure to a range of spore doses. We learn about the parameters of the mathematical model of inhalational anthrax through Bayesian statistical inference methods, making use of previously-published experimental measurements of bacterial loads and protective antigen (PA) levels in New Zealand white rabbits and guinea pigs, together

**Funding:** This work was supported by a UK Engineering and Physical Sciences Research Council (EPSRC) CASE studentship, project reference 2345914, in partnership with Dstl under contract number DSTLX-1000142022 (BW). This work was also supported by an EPSRC Impact Acceleration Account project in collaboration with Dstl, reference number IAA3114 (MLG, GL, JJG, TRL). The funders had no role in study design, data collection and analysis, decision to publish, or preparation of the manuscript.

with dose-response data sets for these animals. We then use a simplified version of the model to simulate the time taken for an individual to show symptoms after exposure and compare with human incubation-period data from the Sverdlovsk 1979 anthrax outbreak.

## Introduction

Anthrax is the infectious disease caused by the bacterium *Bacillus anthracis*. Under adverse conditions, *B. anthracis* forms dormant spores, which are unable to replicate. Inhalational anthrax is initiated by *B. anthracis* spores inhaled by a host. The spores travel through the air passages and some can be deposited in the alveoli of the lungs. The generally-accepted model of inhalational anthrax infection is the Trojan horse model [1], which assumes that spores are engulfed by alveolar phagocytes, such as macrophages [2] or dendritic cells [3–5]. Then, once the spores have been phagocytosed, they germinate inside the host cell. If the resulting bacteria survive the microbicidal defences of the phagocyte, they can begin to replicate, forming long chains of vegetative bacterial cells [6]. The infected phagocytes migrate to the lymph nodes that drain the airways. Here, the intracellular bacteria are released into the extracellular environment when the host cell ruptures and dies. The rupture of infected phagocytes is thought to be primarily due to the physical stresses put on the host cell membrane from the chains of vegetative bacteria growing inside it. Once an infected host cell ruptures, the extracellular bacterial chains continue to multiply. This leads to edema and hemorrhage of the mediastinal lymph nodes, and large amounts of fluid in the pleural cavity, which can severely affect breathing [7]. The bacteria also spread into the bloodstream and other organs to establish a systemic infection [8].

One of the characteristic virulence factors that contributes to the pathogenic success of *B. anthracis* is the production of three proteins that are collectively termed anthrax toxin: protective antigen (PA), oedema factor (EF), and lethal factor (LF). EF and LF are the active components that have an effect on cells by interfering with their functions. However, on their own, EF and LF are non-toxic, since they are not able to enter cells without PA molecules being present. PA binds to receptors on host cells and is cleaved by a host protease called furin. This creates a binding site for either EF or LF. Thus, EF in combination with PA forms the oedema toxin, and LF in combination with PA forms the lethal toxin. These two toxins cause different cellular responses and are essential factors for the survival of *B. anthracis* in an infected host. Lethal toxin disrupts cell signalling pathways of macrophages and some other cells, leading to cell death. On the other hand, oedema toxin inhibits the phagocytosis of bacteria by neutrophils [9]. In some cell types, oedema toxin also increases the levels of cyclic adenosine monophosphate, which is a chemical messenger that plays a major role in controlling many intracellular processes. During systemic infection, the toxins are distributed via the bloodstream to different organs, where they affect the functioning of organs such as the spleen, lymph nodes, liver, kidney, heart, and brain. Together, the anthrax toxins cause disregulation, immune pathology, and death.

Inhalational anthrax is usually fatal if not rapidly detected and treated [10]. Although some cases of inhalational anthrax can be resolved with antibiotic treatment, the fatality rates with treatment are still rather high, since (often times) treatment is not started early enough after exposure and symptom onset. Additionally, *B. anthracis* spores can be produced and preserved, making inhalational anthrax a potential biological terrorism threat [11]. We therefore aim to develop a mathematical within-host model of inhalational anthrax to improve our

understanding of the early progression of the infection, and to help quantify key infection mechanisms.

Several types of mathematical models have previously been developed to study inhalational anthrax. Statistical dose-response models can be used to describe the probability of response for different initial inhaled doses of spores, at the population level. However, these models provide limited information about within-host disease mechanisms, or the timescale of pathogenesis. A standard approach in dose-response assessment is the use of single-hit models. These models assume that when an individual is infected with a pathogen, the organisms act independently in the host so that the probability that any one organism in the initial dose produces an eventual infection is independent of the size of the dose, and the probability of infection is equivalent to the probability that at least one of the organisms in the initial dose will lead to an infection. An example of these single-hit approaches is the exponential dose-response model, which involves a single parameter to denote the probability that an individual organism will produce a response. The competing-risks model for inhalational anthrax was proposed by Brookmeyer et al. [12] as an extension of the exponential dose-response model. This is also a single-hit model, but allows some insight into disease mechanisms since it includes parameters such as the germination rate of spores and the rate at which spores are cleared from the lungs. However, it is still a rather simple model. The hypothesis of the competing-risks model for anthrax is that if a single spore survives ingestion by a macrophage and successfully germinates without being cleared, then the resulting bacterium will be certain to establish an infection. However, some immune cells may be able to kill vegetative *B. anthracis* bacteria [13]. This implies that once a spore in an infected cell has germinated and bacteria are released, it is not guaranteed that these bacteria will survive and cause infection. Another simplification used in the competing-risks model is that the times until germination or clearance of a spore are both assumed to be exponentially distributed. However, this might not be the case. For instance, in the intracellular model presented in [14], as well as the one by Pantha et al. [15], the consideration of the newly germinated bacterium means that the total germination-maturation time is non-exponential.

Deterministic models have also been developed for inhalational anthrax, consisting of differential equations to model variables of the within-host disease dynamics. These models can capture and quantify key biological mechanisms. Some deterministic models are extremely complicated, for example the model by Day et al. [16] includes variables for spores, bacteria, toxin, different types of immune cells, and antibiotics. It models the dynamics in two compartments representing the lungs and the lymph nodes. Model parameters are not calibrated with specific data, but the model aims to provide a qualitative description of the disease in humans for sensible parameter choices. On the other hand, Gutting [17] has developed a simple model to describe the bacterial kinetics in a rabbit model of inhalational anthrax, and used various data sets to calibrate model parameters. However, this model only describes the amount of bacteria within the lung airways and the rabbit body as a whole, rather than dividing the body into multiple compartments or incorporating the dynamics of the anthrax toxin.

Here, a novel model of the within-host dynamics of inhalational anthrax is presented, which could be considered to fall between the one by Day et al. [16] and Gutting [17] in terms of model complexity. The stochastic nature of the within-host model presented here allows it to capture the intrinsic randomness in infection dynamics. For example, the model considers inter-phagocyte variability in the amount of bacteria released from an infected phagocyte by incorporating the rupture-size distribution from a previous intracellular model [14] (developed by our group). Heterogeneity of the rupture size has been shown to be important in a similar model for the pathogen *Francisella tularensis* [18], but is something that has

not been considered in previous within-host models of inhalational anthrax. The stochasticity is particularly important when modelling the outcome of infections initiated by a relatively small number of spores deposited in the lungs. The within-host model also incorporates the dynamics of the anthrax toxin during infection, since this is a key component of anthrax pathogenesis and will also be important for modelling anti-PA treatments in future work. The role of toxins was included in the model by Day et al. [16], but the toxin level was modelled as a number between 0 and 1, with arbitrary units. Here the toxin parameters are instead defined in biologically-relevant units and calibrated with available *in vivo* data to enable a more detailed description of toxin dynamics.

Parameter calibration of the within-host model is performed by comparing realisations of the stochastic model to *in vivo* data from rabbit and guinea-pig infections, as well as linking the model to dose-response data for these animal species in order to ensure that the model reliably describes the probability of infection following exposure to different doses of spores. Building a model that offers a realistic mechanistic description of different animal responses to the inhalation of *B. anthracis* spores is an important step towards the ultimate aim of developing a model that can predict the dynamics of human infection with and without treatments. As well as enabling computation of the infection probability, the stochastic approach also allows quantification of the incubation period distribution (*i.e.*, the time between exposure and symptoms onset). We illustrate this by fitting a simplified version of the within-host model to human incubation-period data from the Sverdlovsk 1979 anthrax outbreak [19]. An analytic approximation has also been derived, allowing rapid evaluation of the incubation-period distribution, given dose.

## Results

### Within-host model for animals

The mathematical within-host model of inhalational anthrax is depicted in Fig 1 and described in detail in the Methods section. The model assumes that each spore in the initial inhaled dose of $D$ spores has probability $\hat{\phi}$ to be deposited in the lung airways. These spores are then removed from the airways of the lungs at rate $\rho$, which encompasses three main removal processes: (i) clearance of spores by ciliated epithelial cells that beat and propel material up the airways to be expelled, (ii) phagocytosis of spores leading to killing of the spore or rupture in a tissue not conducive to bacterial growth, and (iii) phagocytosis leading to survival of the spore, trafficking to the draining lymph nodes of the lungs, and rupture of the cell releasing some positive number of bacteria into the lymph nodes. Only in case (iii) will the spore have a chance to cause an infection. Therefore we define $q$ to be the probability that a given spore in the lungs will successfully infect a phagocyte and lead to some positive number of bacteria being released into the lymph nodes; that is, case (iii) (successful infection of a phagocyte) occurs with probability $q$, and case (i) or (ii) (clearance) occurs with probability $1-q$.

The variable $P$ represents infected phagocytes (each assumed to have phagocytosed one spore only) that will migrate to the lymph nodes and rupture, releasing some bacteria. The time taken for an infected phagocyte to migrate to the lymph nodes and rupture, releasing some positive number of bacteria, is assumed to be Erlang-distributed. The shape parameter of this distribution is chosen to be $n = 3$, which is informed by an Erlang approximation to the rupture-time distribution of the intracellular model calibrated in [14]. In our within-host model, the unit of measurement for the amount of bacteria is colony-forming units (CFU) because this is the unit of the data to which we calibrate the model. *B. anthracis* forms very long chains of bacteria, where one chain is likely to produce one CFU. The number of CFU

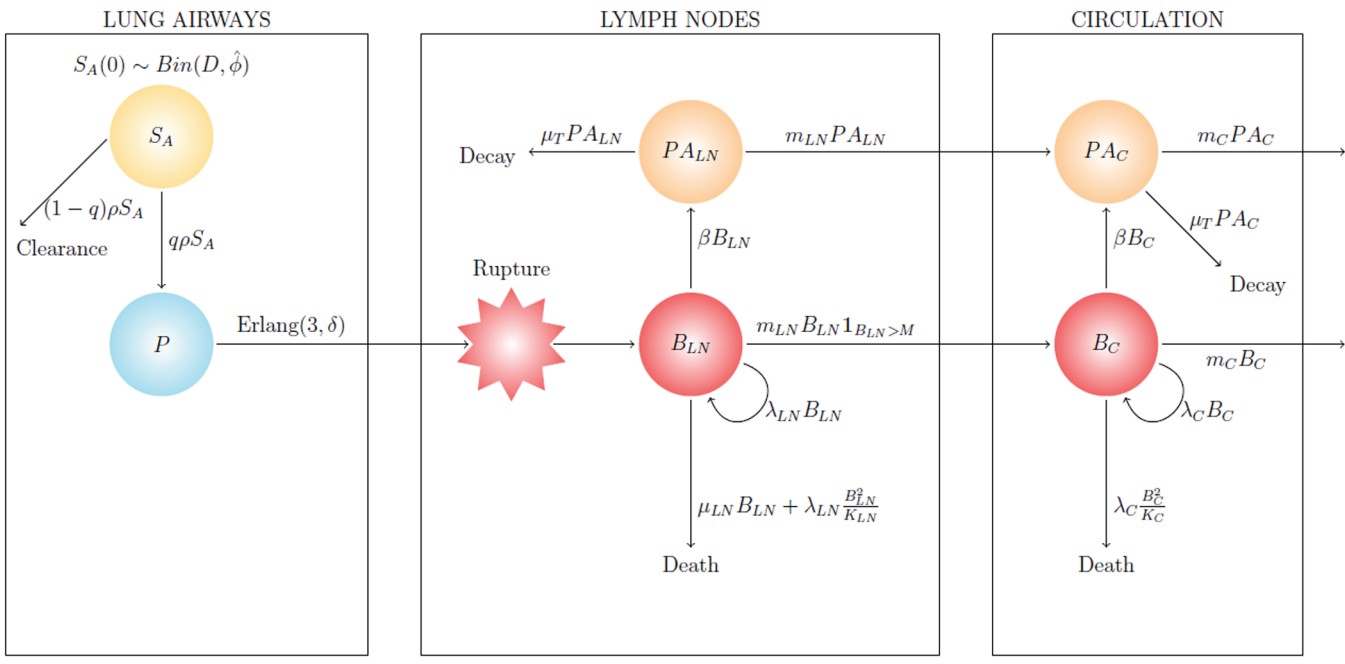

**Fig 1. Diagram of the within-host model of inhalational anthrax.** See the Methods section for a detailed description of the model.

released from an infected phagocyte (the rupture size) is assumed to follow a geometric distribution with mean $R = 1.6$ [14]. Note that many individual bacteria might emerge from a single phagocyte, but these will have formed chains of bacteria, so the number of CFU (chains) released is likely to be much lower than the number of bacteria. We expand on this in the Discussion.

In the lymph nodes (LN), the extracellular bacteria replicate at rate $\lambda_{LN}$ per CFU, while death of extracellular bacteria in the lymph nodes is assumed to occur at rate $\mu_{LN}$ per CFU due to the action of the host immune system. This single term accounts for multiple mechanisms of extracellular bacterial death, such as from neutrophils, macrophages, dendritic cells, lymphocytes, and soluble components of the innate immune system [13,16,20]. Furthermore, a non-linear term in the death rate accounts for competition for resources between extracellular bacteria, such that bacterial proliferation is modelled by logistic growth, with carrying capacity $K_{LN}$. Migration of extracellular bacteria from the lymph nodes to circulation in the blood vessels occurs at rate $m_{LN}B_{LN}$ if the amount of bacteria in the lymph nodes is greater than the migration threshold, $M$ (see the Methods section for a discussion on this). In the circulation, proliferation of bacteria is modelled by logistic growth, with replication rate $\lambda_C$ and carrying capacity $K_C$. Bacteria circulating in the blood are removed into organs such as the liver and spleen at rate $m_C$. Free protective antigen (PA), produced by bacteria in the lymph nodes and circulation at rate $\beta$, is removed due to degradation or host cell binding at rate $\mu_T$, and moves between compartments in a similar way to bacteria. Table 1 provides a list of the parameters in the within-host model.

If an individual inhales a dose of *B. anthracis* spores, it is possible that their innate immune system will be capable of clearing the pathogen, without resulting in a detectable infection. However, in some cases the infection cannot be contained and will become established. The

**Table 1**. **Descriptions (and units) of model parameters, organised by model compartment.**

| | Parameter | Unit | Description |
|---|---|---|---|
| Lung airways | $\hat{\phi}$ | - | Probability that a given inhaled spore will become deposited in the lungs |
| | $\rho$ | $h^{-1}$ | Rate of removal of spores from the lung airways |
| | $q$ | - | Probability that a given spore in the lungs will lead to a rupture event in LN |
| | $\delta$ | $h^{-1}$ | Rate parameter of Erlang-distributed time for migration to LN and rupture |
| Lymph nodes (LN) | $R$ | CFU | Mean of the geometric distribution of the rupture size |
| | $\lambda_{LN}$ | $h^{-1}$ | Logistic growth rate of extracellular bacteria in the lymph nodes |
| | $\mu_{LN}$ | $h^{-1}$ | Killing rate of extracellular bacteria in the lymph nodes |
| | $K_{LN}$ | CFU | Carrying capacity for logistic growth of extracellular bacteria in LN |
| | $M$ | CFU | Threshold amount of bacteria needed in LN before migration into circulation |
| | $m_{LN}$ | $h^{-1}$ | Migration rate of extracellular bacteria and PA from LN into circulation |
| Circulation (C) | $\lambda_C$ | $h^{-1}$ | Logistic growth rate of extracellular bacteria in the circulation compartment |
| | $K_C$ | CFU | Carrying capacity for logistic growth of extracellular bacteria in the circulation |
| | $m_C$ | $h^{-1}$ | Migration rate of extracellular bacteria and PA out of circulation |
| Protective antigen (PA) | $\beta$ | ng $(CFU \cdot h)^{-1}$ | Production rate of PA by extracellular bacteria |
| | $\mu_T$ | $h^{-1}$ | Removal rate of PA due to natural degradation and binding to cells |

within-host model can be used to investigate the probability that infection becomes established, given a known initial dose of inhaled spores. The calculation of this probability of infection is shown in the Methods section, resulting in an expression for the per spore probability of infection, $r$, in Eq (5), and the exponential dose-response model shown in Eq (6). The exponential dose-response model in Eq (6) is widely used in microbial risk assessment, and has been previously used many times to establish dose-response curves for inhalational anthrax. For example, Gutting et al. [21] have fitted the exponential dose-response model to dose-response data sets for different animal species, in order to estimate the parameter $r$ for each species. The novelty of our approach is that Eq (5) allows for a mechanistic interpretation of $r$ in terms of the early within-host infection dynamics in Fig 1.

Here, the model in Fig 1 is calibrated using data from studies of New Zealand white rabbits [21,22] and guinea pigs [21,23]. For each species, the model is fitted to measurements of bacterial CFU from animals that were exposed to a high dose of spores, as well as mortality rates from dose-response data sets, simultaneously. In this way, it is possible to ensure that the model accurately describes both the *in vivo* dynamics of infection, and the probability of infection after exposure to various doses of spores. For guinea pigs, we also use measurements of PA levels in the blood, which allows us to calibrate the model parameters that describe the PA dynamics.

To calibrate the model for rabbit infection, we used the rabbit dose-response data set presented by Gutting et al. [21]. Additionally, we used data from a study by Gutting et al. [22] in which rabbits were infected with the highly virulent Ames strain of *B. anthracis* in order to characterise bacterial dissemination following exposure to a high aerosol dose of spores.

This data set provides measurements of the number of bacterial CFU in the tracheo-bronchial lymph nodes (TBLN) and blood of individual rabbits at times 1, 6, 12, 24, and 36 hours after exposure to a mean inhaled dose of $4.428 \times 10^7$ spores. To calibrate the model for guinea-pig infection, we used the guinea-pig dose-response data set presented by Gutting et al. [21]. Additionally, we used data from a study by Savransky et al. [23] in which guinea pigs were infected with the Ames strain of *B. anthracis* using nose-only aerosol exposure. This data set provides measurements of the number of CFU per ml of blood and the amount of PA (ng/ml) in sera for individual guinea pigs at times 24, 30, 36, 48, and 72 hours after exposure to a dose of $2 \times 10^7$ spores. The details of each data set are provided in the Methods section.

**Rabbit model calibration.** The model in Fig 1 was simultaneously fitted to rabbit dose-response data and *in vivo* numbers of bacteria, using Approximate Bayesian Computation Sequential Monte Carlo (ABC-SMC) [24], in order to estimate the posterior distribution of the parameter vector $(q, \delta, \lambda_{LN}, \mu_{LN}, M, m_{LN}, m_C)$. Other parameters were fixed to the values $\hat{\phi} = 0.092$, $\rho = 0.0735 \text{ h}^{-1}$, $R = 1.6$ CFU, $K_{LN} = 10^9$ CFU, $\lambda_C = 0.17 \text{ h}^{-1}$, and $K_C = 10^{11}$ CFU. Justification for this and details about the ABC-SMC algorithm used are provided in the Methods section. We have not estimated the toxin parameters for rabbits, since the data set used does not contain PA measurements.

Model predictions from this calibration are shown in Fig 2, where the posterior parameter set corresponding to the smallest distance from the data has been used to obtain 20 stochastic model realisations. The plots show the predicted number of CFU in the two different model compartments, along with the *in vivo* observations (dots showing the CFU measurements from individual rabbits, provided in Table 4). The parameter sets in the posterior sample have also been used to obtain predictions of the dose-response curve (see Eqs (5) and (6) in the Methods section). In Fig 3, the pointwise median and 95% credible interval (CI) of the model predictions from all parameter estimates in the posterior sample are compared with the observed response rates in groups of rabbits exposed to different doses. The "response" measured in this data set was mortality, but we assume that established inhalational anthrax will

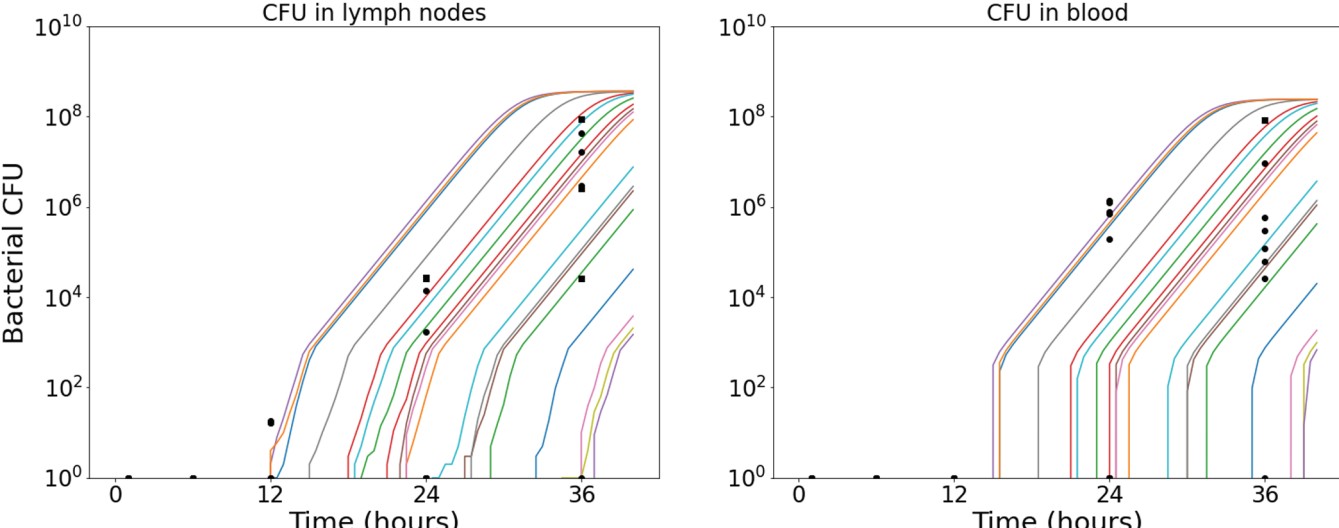

**Fig 2. Stochastic model realisations compared to the CFU data used in the ABC-SMC for the rabbit model calibration.** A sample of 20 model realisations using the "best" posterior parameter set (the one corresponding to the smallest distance from the data in the ABC-SMC) is compared with the CFU loads in the TBLN (left) and blood (right) of individual rabbits, provided in Table 4.

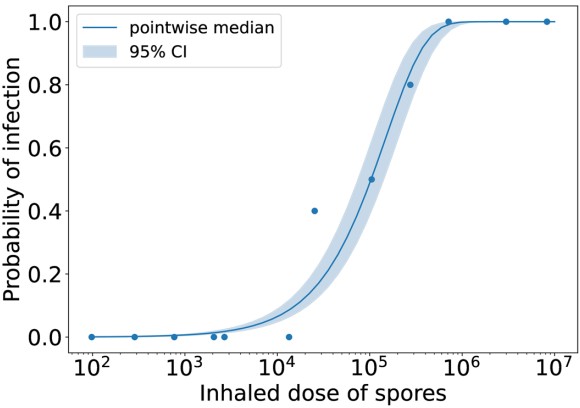

**Fig 3. Posterior predictions of the dose-response curve, compared with the dose-response data used in the ABC-SMC for the rabbit model calibration.** Each parameter set in the posterior sample was used to obtain a prediction from the mechanistic exponential dose-response model given by Eqs (5) and (6). The blue solid line shows the point-wise median of these predictions and the shaded region shows the 95% credible interval. The dots represent the rabbit dose-response data set from Gutting et al. [21], which was used in the calibration.

invariably lead to death in the absence of treatment. Thus, the number of animals that died in the study is assumed to be equivalent to the number that became infected [25].

From the marginal posterior distributions in Fig 4, one can see that it has been possible to significantly constrain most of the rabbit model parameters. One parameter with a particularly narrow posterior distribution is $\delta$, which determines how quickly infected phagocytes migrate from the lungs to the lymph nodes and rupture. Once the first phagocyte has ruptured and released bacteria into the lymph nodes, the dynamics in the lymph nodes begins to be dominated by the extracellular replication of bacteria, and the subsequent rupture events do not have a significant impact on the dynamics of the model. Therefore, the variability between different stochastic realisations of the model is mainly due to the randomness in the time until the first rupture event occurs. If the value of $\delta$ is too small, there is likely to be a long delay between exposure and the time at which bacteria are first detected in the lymph nodes of an individual, which would not be consistent with the rabbits for which bacteria were detected in the lymph nodes at 12 hours. On the other hand, if the value of $\delta$ is too large, the first rupture event will be guaranteed to occur very quickly and this would not be consistent with the one rabbit that did not have a measurable bacterial load at 36 hours. For the posterior median value of $\delta = 0.02$ h$^{-1}$, and assuming the average number of phagocytes that will rupture is 285 (a dose of $4.428 \times 10^7$ spores, $\hat{\phi} = 0.092$, and $q = 7 \times 10^{-5}$), the Erlang(3, $\delta$) distribution of the time for phagocytes to rupture predicts that in 50% of cases, the quickest rupture event will occur within around 13 hours of phagocytosis. This allows the model to capture the observed variability between infection time courses of individual animals in the study by Gutting et al. [22].

The posterior sample of values for the replication rate of extracellular bacteria in the lymph nodes, $\lambda_{LN}$, has a median of 1.49 h$^{-1}$ and a 95% CI of (1.04, 2.53). This is slightly larger than previous estimates for the replication rate of *B. anthracis in vitro*. For example, a median estimate of 0.97 h$^{-1}$ was obtained by fitting a mathematical model of *B. anthracis* growth to data from *in vitro* experiments carried out at Dstl [26]. Elsewhere, Day et al. [16] used an estimate of 0.8 h$^{-1}$ for the growth rate of extracellular bacteria in their model of inhalational anthrax, and an estimate of 0.64 h$^{-1}$ for the growth rate of intracellular *B. anthracis* has been obtained by fitting a mathematical model to *in vitro* macrophage infection data [14]. Thus, our results

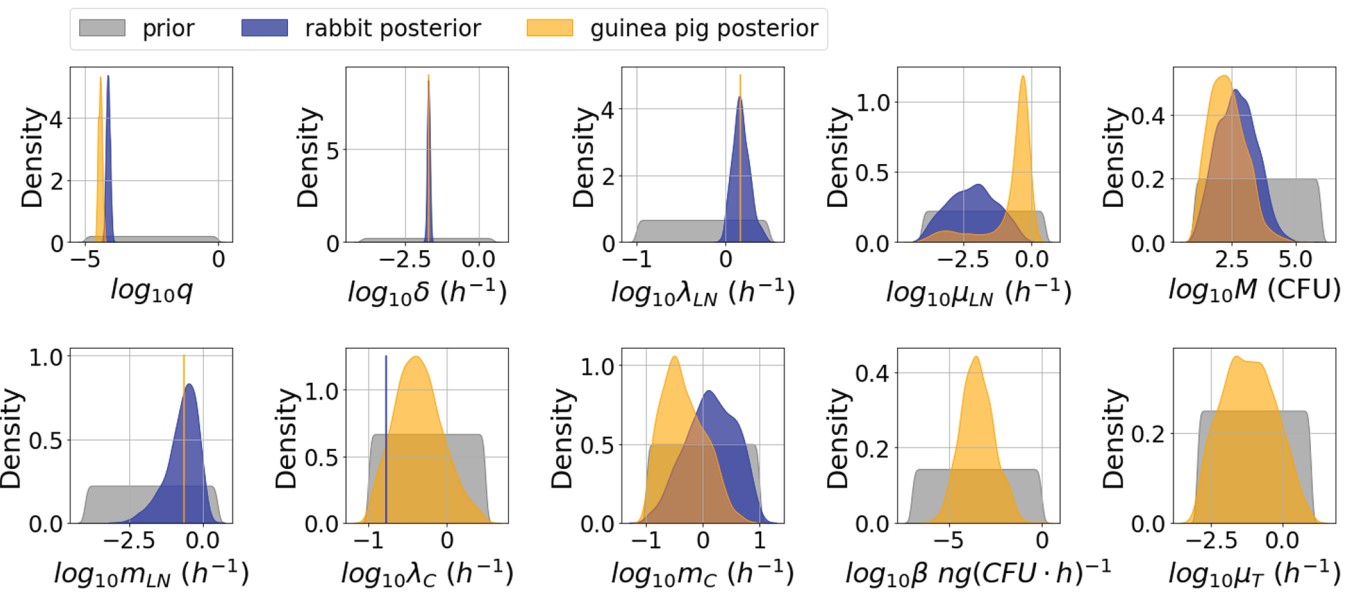

**Fig 4. Posterior distributions for each species.** Kernel density estimates for the prior distribution (grey) and the marginal posterior distribution of each parameter in the rabbit model (blue) and the guinea-pig model (orange). Parameter values that were fixed for the rabbit or guinea-pig model calibration are indicated by vertical blue or orange lines, respectively. For each species, the corresponding posterior distribution was obtained by fitting the mechanistic dose-response curve given by Eqs (5) and (6) to the dose-response data for that species from Gutting et al. [21], and simultaneously fitting the Markov-chain model in Fig 1 to the CFU loads in the TBLN and blood from the rabbit data in Table 4, or the CFU loads and PA amounts in the blood from the guinea-pig data in Table 5.

predict that replication could be faster *in vivo*. However, in our posterior sample, smaller values of $\lambda_{LN}$ correspond to smaller values of the removal rate of bacteria from the blood, $m_C$. Therefore, being able to better determine this removal rate would allow us to more confidently estimate the replication rate of extracellular bacteria in the lymph nodes.

The killing rate of bacteria in the lymph nodes, $\mu_{LN}$, has a fairly wide posterior distribution. However, this parameter is estimated to be small, which is consistent with the competing risks hypothesis that if bacteria are released from a phagocyte, then these will be almost certain to proliferate and cause an infection.

The posterior sample of values for the migration threshold, $M$, representing the number of bacterial CFU that must be present in the lymph nodes before migration into the circulation compartment can occur, has a median of 467 CFU and a 95% CI of $(21, 1.3 \times 10^4)$. This migration threshold helps to explain the delay between observing bacteria in the lymph nodes and in the blood.

**Guinea-pig model calibration.** For guinea pigs, Gutting et al. [21] estimated that the probability of deposition was $\hat{\phi} = 0.3$. We therefore fixed this value of the deposition probability in the guinea-pig model. We consider $\rho = 0.0735$ h$^{-1}$, $R = 1.6$ CFU, $K_{LN} = 10^9$ CFU, and $K_C = 10^{11}$ CFU, as we did for rabbits (see the Methods section for justification).

With only data for the blood compartment, it is difficult to quantitatively estimate the dynamics of infection in the lymph nodes of guinea pigs. Thus, we have aimed to reduce the parameter space by assuming that some parameters determining the dynamics in the lymph nodes would be relatively similar to the ones for rabbits. We have fixed the values of some parameters to those estimated for the rabbit model of infection, while still allowing important differences between the two animal models. In particular, we have fixed the values of $\delta = 0.02$ h$^{-1}$, $\lambda_{LN} = 1.49$ h$^{-1}$, and $m_{LN} = 0.24$ h$^{-1}$ to the median values estimated for rabbits.

We have then used ABC-SMC to estimate the posterior distribution of the parameter vector $(q, \mu_{LN}, M, \lambda_C, m_C, \beta, \mu_T)$ describing guinea-pig infection. Similar to the rabbit model calibration, we have simultaneously fitted the model in Fig 1 to the guinea-pig dose-response data from Gutting et al. [21] and the *in vivo* measurements of bacterial CFU and PA from Savransky et al. [23], provided in Table 5. Model predictions from this calibration are shown in Figs 5 and 6.

Fig 4 and Table 2 compare the marginal posterior distributions of parameters that were estimated separately for rabbits and guinea pigs. The posterior distribution for $\mu_{LN}$ is quite different between the two species. This is because, overall, the data seem to indicate that the growth of bacteria in the blood may be slower in guinea pigs than rabbits. In the model in Fig 1, the overall growth rate of bacteria in the blood depends strongly on the growth rate of bacteria in the lymph nodes. Therefore, the death rate of bacteria in the lymph nodes, $\mu_{LN}$, is estimated to be larger for guinea pigs than rabbits in order to represent slower growth in the lymph nodes and hence capture the slower growth observed in the blood compartment. On the other hand, the value of the migration threshold, $M$, is estimated to be slightly smaller for guinea pigs than rabbits. It could be that $M$ scales between species according to relative body weight and organ volume. The consequence of these two differences in parameter values is that, for each species, the model predicts a similar delay between the time at which bacteria begin replicating in the lymph nodes and the time at which bacteria are first detected in

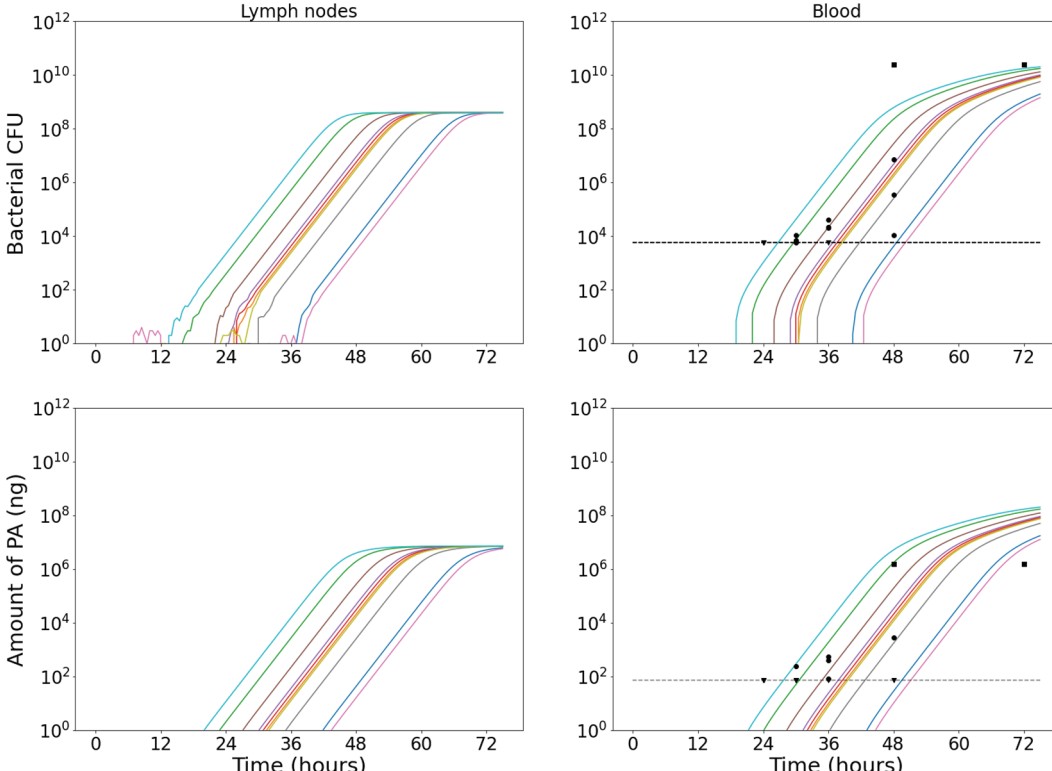

**Fig 5. Stochastic model realisations compared with the CFU and PA data used in the ABC-SMC for the guinea-pig model calibration.** A sample of 10 model realisations using the "best" posterior parameter set (the one corresponding to the smallest distance from the data in the ABC-SMC) is compared with the CFU loads (top row) and PA amounts (bottom row) in the blood of individual guinea-pigs, provided in Table 5.

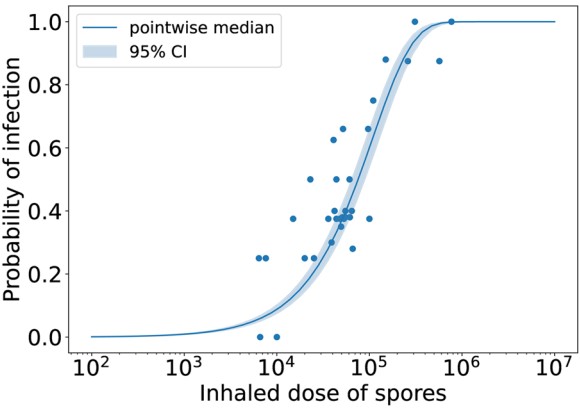

**Fig 6. Posterior predictions of the dose-response curve, compared with the dose-response data used in the ABC-SMC for the guinea-pig model calibration.** Each parameter set in the posterior sample was used to obtain a prediction from the mechanistic exponential dose-response model given by Eqs (5) and (6). The blue solid line shows the pointwise median of these predictions and the shaded region shows the 95% credible interval. The dots represent the guinea-pig dose-response data set from Gutting et al. [21], which was used in the calibration.

**Table 2**. **Medians and 95% credible intervals of the marginal posterior distributions for each parameter and each species.** The values in bold indicate fixed values that were used.

| | Parameter | Unit | Rabbits | Guinea pigs |
|---|---|---|---|---|
| Lung airways | $\hat{\phi}$ | - | **0.092** | **0.3** |
| | $\rho$ | $h^{-1}$ | **0.0735** | **0.0735** |
| | $q$ | - | $7.4\times10^{-5}$ ($5.4\times10^{-5}$, $1.0\times10^{-4}$) | $3.8\times10^{-5}$ ($2.8\times10^{-5}$, $5.2\times10^{-5}$) |
| | $\delta$ | $h^{-1}$ | $2.0\times10^{-2}$ ($1.6\times10^{-2}$, $2.4\times10^{-2}$) | **0.02** |
| Lymph nodes (LN) | $R$ | CFU | **1.6** | **1.6** |
| | $\lambda_{LN}$ | $h^{-1}$ | 1.49 (1.04, 2.53) | **1.49** |
| | $\mu_{LN}$ | $h^{-1}$ | $7.3\times10^{-3}$ ($1.7\times10^{-4}$, $2.7\times10^{-1}$) | $4.1\times10^{-1}$ ($3.4\times10^{-4}$, $7.3\times10^{-1}$) |
| | $K_{LN}$ | CFU | $\mathbf{10^9}$ | $\mathbf{10^9}$ |
| | $M$ | CFU | 467 (21, $1.3\times10^4$) | 182 (16, $6.7\times10^3$) |
| | $m_{LN}$ | $h^{-1}$ | $2.4\times10^{-1}$ ($9.5\times10^{-3}$, 1.1) | **0.24** |
| Circulation (C) | $\lambda_C$ | $h^{-1}$ | **0.17** | $4.2\times10^{-1}$ ($1.2\times10^{-1}$, 1.9) |
| | $K_C$ | CFU | $\mathbf{10^{11}}$ | $\mathbf{10^{11}}$ |
| | $m_C$ | $h^{-1}$ | 1.4 ($1.9\times10^{-1}$, 7.1) | $4.0\times10^{-1}$ ($1.2\times10^{-1}$, 2.7) |
| Protective antigen (PA) | $\beta$ | ng $(CFU \cdot h)^{-1}$ | N/A | $3.1\times10^{-4}$ ($7.7\times10^{-6}$, $2.3\times10^{-2}$) |
| | $\mu_T$ | $h^{-1}$ | N/A | $6.1\times10^{-2}$ ($1.7\times10^{-3}$, 3.4) |

the blood. That is, the net bacterial growth rate in the lymph nodes is estimated to be smaller for guinea pigs, but the size that the bacterial population must reach before it starts to enter the blood is also lower. However, for guinea pigs, since the death rate of extracellular bacteria is closer to the replication rate, there is a larger probability that a small number of bacteria released during a rupture event will die before managing to establish an exponentially-growing population of bacteria (this can be seen in a couple of the realisations shown in Fig 5). Hence, on average more rupture events are needed to establish an infection, which can delay the infection slightly.

The probability, $q$, that a deposited spore in the lungs leads to a phagocyte rupture event, during which bacteria are released in the lymph nodes, is estimated to be extremely small. In particular, for rabbits, $q$ is estimated to be around $7 \times 10^{-5}$, and for guinea pigs it is estimated to be around $4 \times 10^{-5}$. These small values of $q$ help explain the relatively high $ID_{50}$ for anthrax that has been observed for these species. Furthermore, the probability $q$ is estimated

to be slightly smaller for guinea pigs than rabbits. This is expected since a higher fraction of inhaled spores is likely to be deposited in the lungs of guinea pigs than rabbits [21], yet the dose-response data sets show that the probability of infection for a given dose of spores is similar between rabbits and guinea pigs. This implies that, in guinea pigs, fewer of the deposited spores will lead to bacteria in the lymph nodes. Biologically, this could be due to a difference between New Zealand white rabbits and guinea pigs in their ability to clear spores from the lungs; for example, due to a difference in the density of macrophages lining the lungs, the speed at which these macrophages phagocytose spores, or the ability for these macrophages to kill intracellular *B. anthracis* once phagocytosed. Each of these possibilities would affect the probability of rupture in the lymph nodes, given that a spore has been deposited in the lungs. On the other hand, the difference in the probability $q$ could be due to a difference in the clearance of spores by the ciliated epithelial cells. However, there is evidence that physical clearance rates in the lungs are broadly similar between rabbits and guinea pigs [27]. Therefore, we believe that species differences in the probability of rupture events in the lymph nodes are likely to be due to differences in antimicrobial activity.

Since the data set from Savransky et al. [23] includes PA levels in the blood of guinea pigs, this has allowed us to calibrate the model parameters corresponding to PA dynamics, which we could not do for the rabbit model. For the production rate of PA, $\beta$, the posterior distribution in Fig 4 has a median of $3.1 \times 10^{-4}$ ng $(\text{CFU} \cdot \text{h})^{-1}$ and a 95% CI of $(7.7 \times 10^{-6}, 2.3 \times 10^{-2})$. This seems consistent with the results in [26] from calibrating a model with *in vitro* data sets from Zai et al. [28] and Charlton et al. [29], corresponding to median estimates for $\beta$ of $2 \times 10^{-5}$, $10^{-4}$, and $3 \times 10^{-2}$ ng $(\text{CFU} \cdot \text{h})^{-1}$.

## Towards modelling human incubation periods

We illustrate the possible applicability of our modelling approach to human inhalational anthrax by considering a simpler version of the model (see Fig 7). Here we have combined the lymph nodes and circulation compartments from the model in Fig 1 into a single compartment, since we have very little human data from which to infer parameter values such as the migration rates, migration threshold, and bacterial proliferation rates in different compartments. Furthermore, we do not think it would be realistic to make use of the parameter values obtained from the calibration with rabbit or guinea-pig data to inform these inter-organ infection dynamics for humans. Details of how the human model is simulated are provided in the Methods section.

We have performed parameter inference via ABC-SMC, using incubation period data from an anthrax outbreak that occurred in Sverdlovsk, Russia in 1979, when *B. anthracis* spores were accidentally released from a military microbiology facility on April 2. The dates of symptom onset for 30 autopsy-confirmed anthrax victims have been reported by Abramova et al. [19]. To obtain a value for the number of days between exposure and symptom onset for each of these individuals, we assume that all exposures to the spores occurred on April 2 [12]. Examples of symptoms reported by these individuals include fever, dyspnea, cough, headache, vomiting, chills, weakness, abdominal pain, and chest pain. Therefore, we assume that "symptom onset" in this data set means the first time that one of the above symptoms was observed (*i.e.*, prodromal symptoms onset). We note that a limitation of these data is that the disease course of some patients may have been impacted by taking tetracycline as a form of post-exposure prophylaxis, but with an unknown start date and unknown duration [30]. Histograms of the incubation period data are shown in Fig 8, and the model fits to the data in Fig 9. We assumed that the doses of spores to which the individuals were exposed follow the distribution estimated by Wilkening for their "Model D" [31, Fig 11].

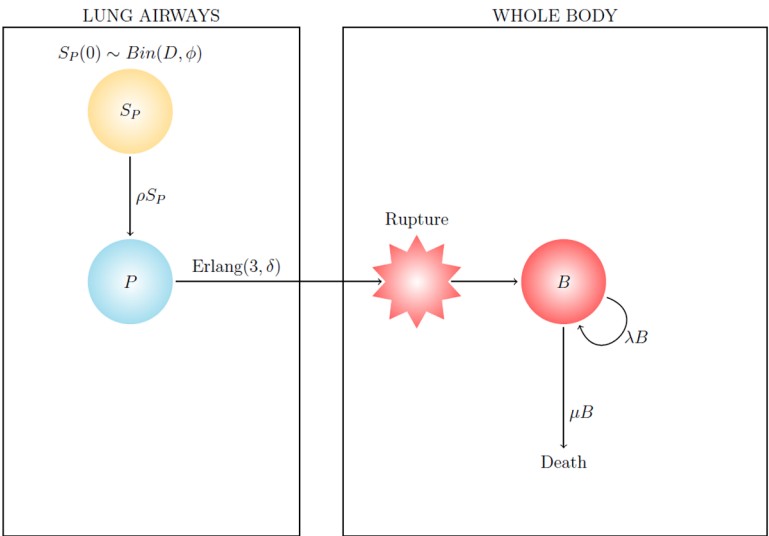

**Fig 7. Diagram of the simplified within-host model of inhalational anthrax for humans.** See the Methods section for a description of how the model is used to simulate individual incubation periods.

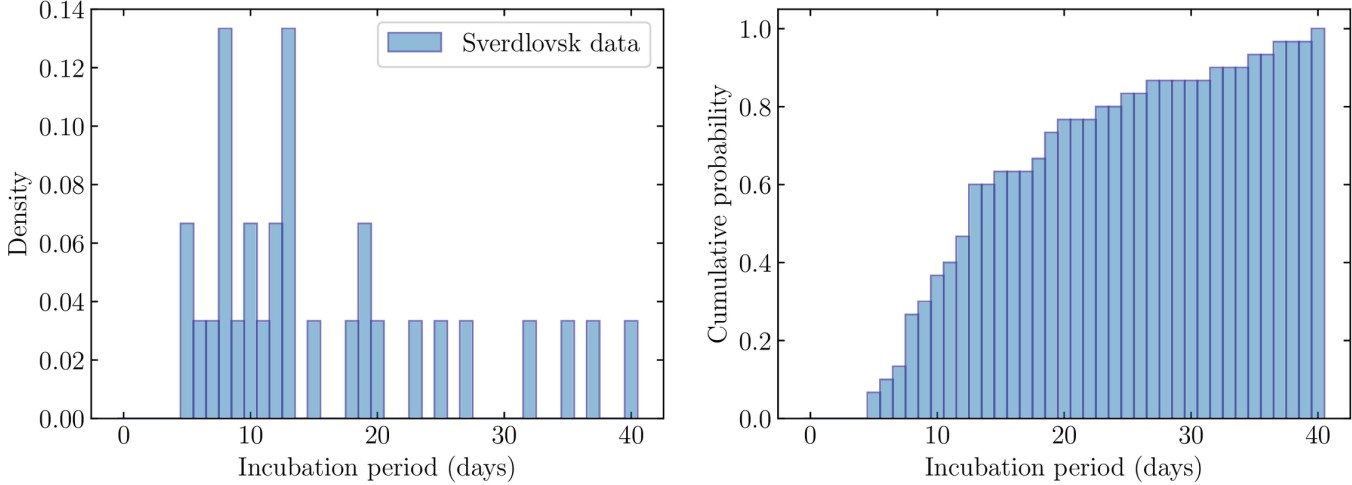

**Fig 8. Human incubation period data set from the Sverdlovsk outbreak [19].** Left: Histogram of the number of days to symptom onset for the 30 individuals. Right: Cumulative distribution of number of days to symptom onset.

Prior to the parameter inference, as in the rabbit and guinea-pig models, the mean rupture size was set to $R = 1.6$ CFU. We then obtained a posterior distribution for the parameter vector $(\rho, \delta, \sigma, p)$, where we have used a parameter transformation of $\sigma = \lambda - \mu$ and $p = \frac{\mu}{\mu + \lambda}$. That is, $\sigma$ represents the net growth rate of extracellular bacteria in the human body, and $p$ represents the probability for extracellular bacteria to be killed by host immune cells before replicating. To transform back to the original model parameters,

$$\lambda = \frac{\sigma(1-p)}{1-2p}, \qquad \mu = \frac{\sigma p}{1-2p}. \tag{1}$$

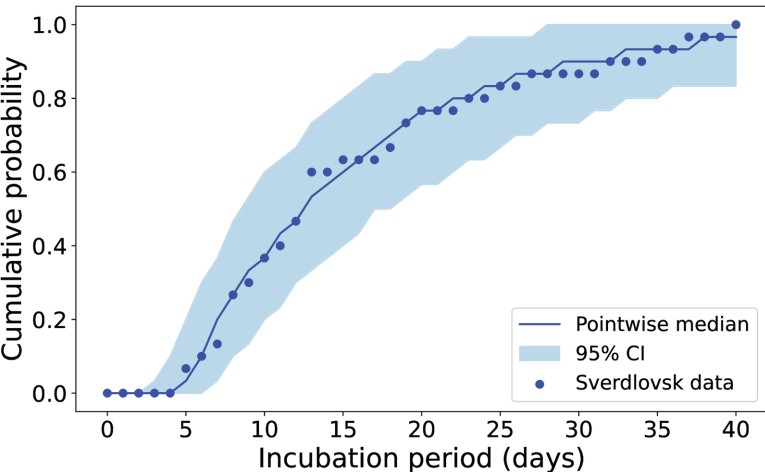

**Fig 9. Pointwise median and 95% credible interval of the model fits to the human incubation period data [19].** For each parameter set in the posterior sample (see Fig 10), a sample of 30 finite incubation periods was obtained from stochastic model simulations. We plot here the median and 95% CI (across all posterior parameter sets) of the cumulative daily fractions of these simulated incubation periods.

While finding parameter values for which the model is able to describe the human incubation period data, we imposed a constraint on the parameters so that the model produces a consistent dose-response curve. In particular, we fixed the value of $r$, which determines the exponential dose-response relationship in Eq (6). Wilkening [31,32] has previously modelled the incubation period distribution for human inhalational anthrax using data from the Sverdlovsk outbreak. They assumed an $ID_{50}$ (the dose at which there is a 50% chance of infection) of 8,600 spores, since this is a nominal value for highly virulent *B. anthracis* strains. Thus, we set the probability that a single spore will produce a response to $r = 8.06 \times 10^{-5}$ (equivalent to $ID_{50} = 8,600$). Then, using Eq (5), we set the probability that a given inhaled spore is deposited in the lungs, phagocytosed, and leads to a rupture event, to be

$$\phi = \hat{\phi}q = r\left(1 + \frac{p}{R(1 - 2p)}\right). \tag{2}$$

For larger values of $p$ (meaning death of extracellular bacteria is more likely), this probability, $\phi$, increases, since more rupture events will be required in order to have the same probability of infection establishment for a given initial dose of spores.

During the ABC-SMC algorithm, values for the parameters ($\rho, \delta, \sigma, p$) were sampled from the log-uniform prior distributions shown in grey in Fig 10. For each sampled set of parameter values, we compared the model with the observed incubation period data by using the model to simulate finite incubation periods for 30 individuals (same as the number of observed incubation times in the data set in Fig 8). That is, for each individual, we sampled an exposed dose from the distribution estimated by Wilkening for "Model D" [31, Fig 11] and simulated the model as many times as necessary to obtain a finite incubation period. We then calculated the Euclidean distance between the cumulative daily fractions of the sets of simulated and observed incubation times, for days 0 to 40.

The marginal posterior distributions obtained via ABC-SMC are shown in green in Fig 10. The distribution of the phagocytosis rate, $\rho$, is very narrow. The median value of $\rho = 0.004\ \text{h}^{-1}$

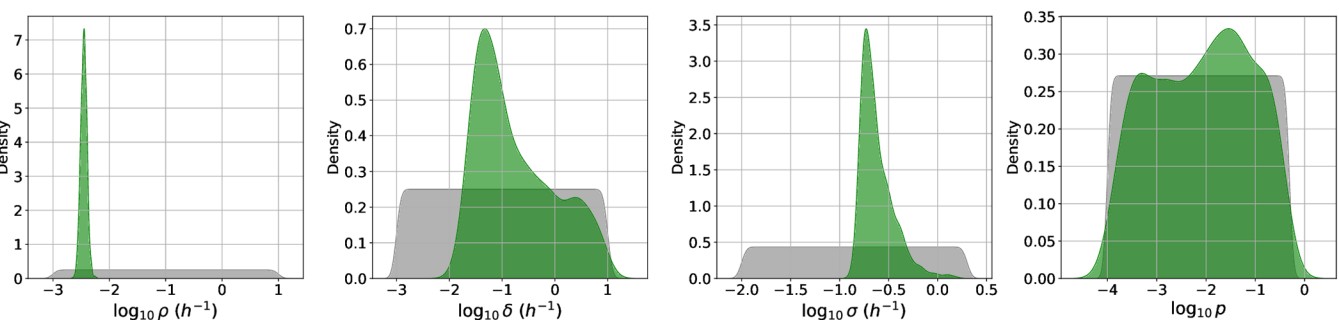

**Fig 10. Posterior distribution for the human model.** Kernel density estimates for the prior distribution (grey) and the marginal posterior distribution (green) of each parameter in the human model. This posterior distribution was obtained by fitting the simple model in Fig 7 to the Sverdlovsk outbreak incubation periods data set in Fig 8.

is consistent with a previous estimate of 0.07 day$^{-1}$ for the clearance rate of spores from the lungs, which was based on data from examination of the lungs of non-human primates at varying times after inhalation [12,25,33].

From Fig 10, we learn that the net bacterial growth rate, $\sigma$, is likely to be between around 0.1 and 1.5 h$^{-1}$, but we are not able to constrain $p$. That is, given a net bacterial growth rate within this range, the human incubation period data can be captured with a wide range of values of the bacterial death rate, $\mu$ (see Fig 11).

Using Eq (2), we can transform the posterior distribution for $p$ into a distribution for $\phi$, which is the probability that an inhaled spore leads to a phagocyte rupture event during which bacteria are released in the lymph nodes. This is shown in Fig 11. The value of $\phi$ is estimated to be larger for the human model than it was for rabbits and guinea pigs. This is because we have assumed that humans are more sensitive to infection by *B. anthracis* (*i.e.*, the ID$_{50}$ is assumed to be lower for humans). In other words, the value we used for the parameter $r$ is higher than the one estimated for rabbits and guinea pigs. Therefore, we are making the assumption that humans have a higher probability of infection for a given dose of inhaled spores. Still, the area of smaller $\phi$ in the posterior distribution in Fig 11, corresponding to smaller values of $p$ and $\mu$, is likely to be the most realistic.

Recall that $\phi$ is defined as the product of the model parameters $\hat{\phi}$ and $q$. The deposition probability of inhaled spores in the lungs, $\hat{\phi}$, depends on the properties of the aerosol inhaled, such as particle size. Thus, if these properties were known, an estimate for $\hat{\phi}$ could be obtained from the Multiple-Path Particle Dosimetry (MPPD) software, which is a computational model that can be used for estimating the deposition and clearance of aerosols in the respiratory tract of humans and various laboratory animal species [34]. This could then be used to recover a posterior distribution for $q$ from the posterior distribution of $\phi$.

In Fig 12 we observe a correlation between $\delta$ and $\sigma$, since if rupture occurs earlier, then the net growth rate of extracellular bacteria needs to be slower in order to explain the incubation period data.

**Approximating the dose-dependent incubation period distribution.** An analytic approximation for the incubation period distribution of our simple human model can be derived by assuming that all initially inhaled spores are independent, and that symptom onset occurs when at least one of these spores has given rise to a population of $N$ extracellular bacteria. In reality, multiple spores could contribute to the bacterial population reaching size $N$, but the assumption of independence is valid for low deposited doses since the probability of enough host cells rupturing within a sufficiently short time window to substantially affect

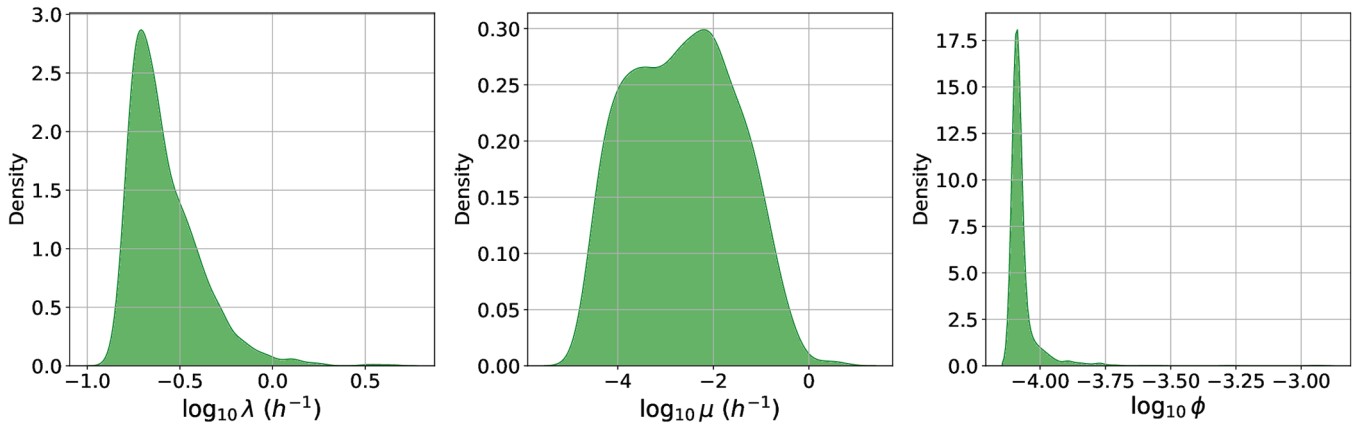

**Fig 11. Kernel density estimates for the marginal posterior distributions of $\lambda$, $\mu$, and $\phi$.** These are obtained by transformations of the posterior distribution for $\sigma$ and $p$, using Eqs (1) and (2).

**Fig 12. Scatter plots showing the relationships between pairs of parameters in the posterior sample for the human model.**

the expected delay is assumed to be negligible [25]. For a Poisson-distributed inhaled dose of spores with mean $D$, and assuming independence between these spores, the probability that symptom onset occurs before time $t$ (*i.e.*, at least one of the initial inhaled spores has given rise to a population of $N$ extracellular bacteria by time $t$) is

$$F(D,t) = 1 - \exp(-DF(1,t)),$$

where $F(1,t)$ is the probability that a single inhaled spore will lead to symptom onset before time $t$.

The probability, $F(1,t)$, that a single inhaled spore will lead to symptom onset before time $t$, is the product of the probability that a single inhaled spore will establish an infection (the probability $r$, given by Eq (5)), and the probability that symptom onset occurs before time $t$, conditioned on infection establishment. Given that a single spore will establish an infection, the time until symptom onset depends on the time taken for the spore to be phagocytosed, the time taken for the infected phagocyte to rupture and release bacteria, and the time taken for this population of bacteria to reach size $N$. To derive an approximation for $F(1,t)$, we use a fixed time, $T$, to approximate the delay between a rupture event and the time at which the extracellular bacterial population reaches the threshold, $N$. The fixed time, $T$, is defined as the time taken for the mean of the bacterial population (represented by a stochastic birth-and-death process) to reach size $N$, starting from a single CFU, and with net growth rate $\sigma$. That is,

$$T = \frac{\log N}{\sigma}.$$

Then, the probability that a single inhaled spore will lead to symptom onset before time $t$ is

$$F(1,t) = \begin{cases} 0 & t \le T, \\ rF_R(t-T) & t > T, \end{cases}$$

where $F_R(t)$ is the probability that the single inhaled spore will lead to a rupture event before time $t$ (given that this spore will eventually lead to an established infection). That is,

$$F_R(t) = \int_0^t \left(1 - e^{-\rho(t-s)}\right) \frac{\delta^n s^{n-1} e^{-\delta s}}{(n-1)!} \, ds$$

$$= 1 - e^{-\rho t} \left(\frac{\delta}{\delta - \rho}\right)^n - \sum_{k=0}^{n-1} \frac{1}{k!} e^{-\delta t} (\delta t)^k \left(1 - \left(\frac{\delta}{\delta - \rho}\right)^{n-k}\right),$$

which is a convolution between the cumulative distribution function of the exponentially-distributed time until phagocytosis (with rate $\rho$), and the probability density function of the Erlang-distributed time between phagocytosis and rupture (with shape parameter $n = 3$ and rate $\delta$).

Finally, to condition on infection establishment, we divide $F(D,t)$ by the probability of infection for an inhaled dose $D$ (the probability $I(D)$, given by Eq (6)). Thus, for a Poisson-distributed inhaled dose of spores with mean $D$, the cumulative distribution function of the incubation period, conditioned on infection becoming established, can be approximated by

$$F^*(D,t) = \begin{cases} 0 & t \le T, \\ \frac{1}{I(D)} \left(1 - \exp\left(-rDF_R(t-T)\right)\right) & t > T. \end{cases} \tag{3}$$

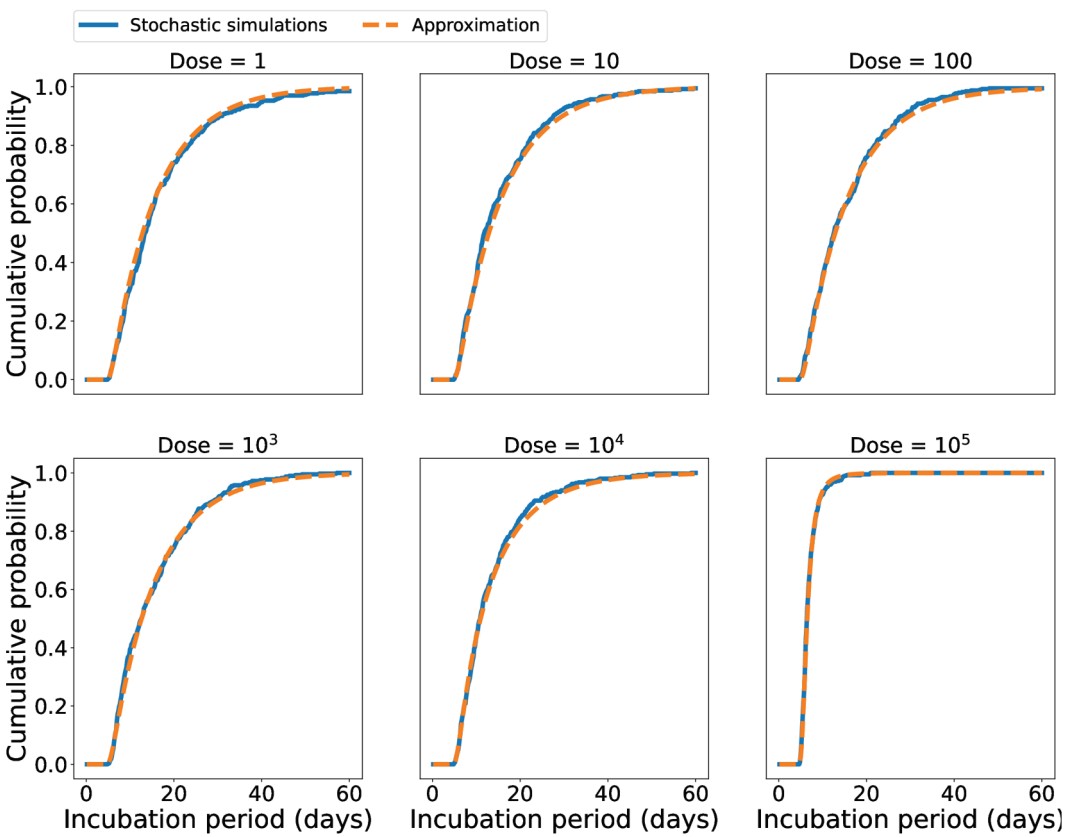

**Fig 13. Comparison between the incubation period distributions obtained by stochastic model simulations and the approximation in Eq (3).** The doses considered are 1, 10, 100, $10^3$, $10^4$, and $10^5$ inhaled spores. For each dose, the blue line was obtained by simulating $4 \times 10^2$ finite incubation periods using the stochastic model. The orange dashed line was obtained by the approximation in Eq (3). The parameter values used were the medians of the marginal posterior distributions in Fig 10: $\rho = 0.004$ h$^{-1}$, $\delta = 0.107$ h$^{-1}$, $\sigma = 0.217$ h$^{-1}$, $p = 0.01$.

Eq (3) is obtained in a similar way to the one obtained by Brookmeyer et al. [12]. However, Brookmeyer et al. assumed a single exponential distribution, representing the time until "germination", whereas we assume separate distributions for the phagocytosis time and the time to host cell rupture, with the latter being Erlang-distributed. We also assume a fixed delay, $T$, between the time at which extracellular bacteria are first present and the time of symptom onset. This is similar to the approach by Toth et al. [25], who estimated this fixed time to be 2.3 days. In comparison, using a threshold of $N = 10^{10}$ CFU [32] and the posterior median net growth rate of $\sigma = 0.217$ h$^{-1}$ (Fig 10), we obtain $T \approx 4.4$ days.

Fig 13 shows some dose-dependent incubation period distributions obtained from simulating the stochastic model, compared with the approximation in Eq (3). We use the median values of $\rho = 0.004$ h$^{-1}$, $\delta = 0.107$ h$^{-1}$, $\sigma = 0.217$ h$^{-1}$, and $p = 0.01$ from the marginal posterior distributions in Fig 10. Using Eq (2), we set the value of $\phi$ so that the model predicts an ID$_{50}$ of 8,600 inhaled spores; with values of $r = 8.06 \times 10^{-5}$, $R = 1.6$, and $p = 0.01$, this results in a value of $\phi = 8.11 \times 10^{-5}$.

## Discussion

We have proposed a mathematical model of the within-host infection dynamics of inhalational anthrax, the disease caused by the bacterium *Bacillus anthracis*. The stochastic approach used allows us to describe the probabilities of different outcomes. For instance, the model incorporates inter-phagocyte variability in rupture size, which is the number of CFU ultimately released from an infected phagocyte. In particular, the rupture size is assumed to follow a geometric distribution, estimated from a previous intracellular model of *B. anthracis* infection [14]. This rupture size probability distribution plays an important role when using the model to determine the probability of infection following inhalation of a dose of spores. Similar to the competing-risks model by Brookmeyer et al. [12], the model that we use here is an extension of the exponential dose-response model in which mechanistic detail has been incorporated in the dose-response parameter *r*. The standard competing-risks model involves parameters for the two competing processes of spore germination and spore clearance, but does not explicitly consider phagocyte rupture or extracellular bacterial dynamics. Here, the formula for the dose-response parameter, *r*, obtained in Eq (5), takes into account the deposition probability of inhaled spores, the stochasticity of rupture events, and the possibility that even if a few bacteria are released from a rupturing cell, infection might not become established if these few bacteria are killed by the host immune system. This novel approach could potentially also be applied to other intracellular pathogens.

In order to calibrate model parameters, stochastic realisations of the model were compared with *in vivo* bacterial counts from rabbits [22] and guinea pigs [23] that had been exposed to a high dose of the virulent Ames strain of *B. anthracis*. At the same time, the probability of infection resulting from the stochastic model was compared with dose-response data sets from rabbits exposed to the Ames strain, and guinea pigs exposed to the M36 strain or Ames strain [21]. Our results show that the single model structure proposed here is able to describe both the *in vivo* dynamics of infection and the probability of response, in two different animal species, with slightly different values for some parameters.

As *B. anthracis* replicates, it forms long chains of bacteria. When quantifying bacteria in units of CFU, the authors of the data that we have used did not describe the steps that would be needed to break these chains. Without such steps, the bacteria in a given chain will not have dissociated from each other, and each chain will form a single colony where it lands on the agar media. Therefore, it is reasonable to assume that we have modelled the number of "infectious units" or "chains", rather than the actual number of bacteria. The relevance of *Bacilli* chains to virulence is not fully understood. However, this process is clearly an adaptation that aids the survival and replicative success of the bacteria; this is evidenced by the known and evolved mechanisms of genetic control of chain formation [35]. It is assumed that these chains contribute to the disease through capillary blockage [36]. Such blockages can make tissues hypoxic (thereby releasing nutrients from dead cells) and prevent immune cells from accessing infected niches. It is conceivable that these chains also contribute to protection against further phagocytic attack (although this protection is driven mainly by toxin and capsule).

Due to lack of data, it has been necessary to use data corresponding to different strains of *B. anthracis* (Sterne, Ames, M36) and different animals to inform different parts of our model. For example, the conditional rupture size distribution was informed by an *in vitro* experiment that used Sterne strain and mice cells [14,37]. We then used this rupture size distribution in the within-host model, which was calibrated with data from rabbits and guinea pigs infected with virulent *B. anthracis* strains. Due to a lack of data for the lymph nodes compartment in

guinea pigs, some of the guinea-pig parameter values were set equal to estimates from the rabbit model. Since the CFU counts suggest an initially slower increase in CFU concentration in the blood of the guinea pigs compared with the rabbit data set, it seems likely that there would also be slower growth of CFU in the lymph nodes of the guinea pigs compared with the rabbit model predictions. However, since we do not have data for the CFU in the lymph nodes of guinea pigs, we set most of the parameters determining the dynamics in this compartment to be equal to their median estimates from the rabbit model. The only one of these parameters that we did not fix for guinea pigs was the death rate of bacteria in the lymph nodes. In practice, this approach is equivalent to specifying that a difference in the bacterial death rate mostly accounts for any differences between the dynamics in the lymph nodes compartments of the two species. This assumption could be relaxed if more data become available.

Although we have followed a Bayesian approach to model calibration, which allows uncertainty in parameter estimates to be quantified, we have not considered the parameter values to vary between individual animals. This is clearly a simplification which could be improved with approaches such as mixed effects modelling, which allow the estimation of probability distributions that describe how parameters vary across a population. However, this approach would be difficult here, since we do not have more than one measurement for each animal. Some parameters may be easier to assign a probability distribution to than others. For example, in a dose-response model for Q fever (a bacterial infection caused by *Coxiella burnetii*), Heppell et al. [38] approximated a distribution for the probability of deposition ($\hat{\phi}$ in our model) using the MPPD model software package [34].

The within-host model incorporates the dynamics of PA, which is the binding component of the anthrax toxin. Since PA is thought to be the limiting toxin component and is the one targeted by the therapeutic antibody, the model only includes variables for the amount of PA, and neglects the other two toxin components, EF and LF. We used measurements of PA levels in the blood of infected guinea pigs to calibrate the parameters describing PA dynamics for this species. However, we did not have data for the PA levels in infected rabbits and therefore, we were not able to estimate values of the corresponding parameters for rabbits. A limitation of the model is that it does not incorporate the effect of the anthrax toxin on bacterial dynamics. In future, further data could be used to inform an additional mechanism in the model to describe how toxin levels inhibit the immune response to *B. anthracis*, and in turn, study the effect of anti-PA treatments.

To ensure that the model is simple enough to be calibrated with available data, while still offering a realistic mechanistic description of the infection, we focus on the number of extracellular spores in the lung airways, and the number of bacterial CFU and amount of PA in the lymph nodes and circulation. In future, if relevant *in vivo* data sets become available to justify the additional complexity, a more detailed spatial structure could be included, with additional compartments representing other organs and tissues that may become colonised during the infection. Furthermore, the model is not currently able to explain observations such as spores detected in lymph nodes of mice [39], since it is assumed that at most one spore is engulfed by each phagocyte in the lungs and that only bacteria are released from rupturing phagocytes in the lymph nodes. However, in future, the model could be extended to include the uptake of multiple spores by individual phagocytes, which may be more likely for high inhaled doses. This would also allow us to consider that cells may release a mixture of spores and bacteria when they rupture, which could explain the detection of spores in lymph nodes. The model could then also be adapted to include the fact that these spores may be engulfed by other phagocytes in the lymph nodes, leading to further rounds of intracellular replication.

Our approach assumes that early infection events (*e.g.* during the first 48 hours post-exposure) are the crucial ones for determining the dose-response probabilities and variability

in the incubation period. Furthermore, the model assumes that killing of extracellular bacteria in the lymph nodes occurs at a constant rate of $\mu_{LN}$ per CFU. However, actions of the host immune system that occur later in infection, which are not included in the current model, may be important for modelling the infection dynamics at later times post exposure. For example, the model predictions by Day et al. [16, Fig 2] suggest that neutrophils may not be crucial in the first 48 hours but could help to control the infection at later times if the bacterial load is relatively low. Thus, our model may provide predictions that are less realistic beyond 48-72 hours post exposure.

To illustrate how the mathematical model of inhalational anthrax developed here could be used to link to human data, we adapted the within-host model by leveraging human incubation period data from the Sverdlovsk outbreak [19]. This data set is useful since the date on which exposure occurred can be reasonably estimated and there have been estimates obtained for the distribution of doses that individuals were exposed to. However, this data set has been excluded from some studies in the literature [40], since it is unknown whether, or for how long, individuals may have taken antibiotics post-exposure.

In future, the within-host model could be linked to Quantitative Microbial Risk Assessment (QMRA) techniques [41] to estimate exposure risks at the population level. This type of approach has previously been applied to pathogens such as the bacterium *F. tularensis* [18] and SARS-CoV-2 [42,43]. Dispersion models that describe the airborne spread of *B. anthracis* spores in indoor or outdoor settings, and predict a distribution of initial doses of spores among an exposed population, can be linked to the outputs of the within-host model (*e.g.* likelihood of symptom onset for a given dose) to predict, in different scenarios, the distribution of the number of casualties and fatalities, and the earliest time of symptom onset. Ultimately, the aim is to predict how many individuals would become infected in different exposure scenarios and on what timescale, to aid in the identification of a time window during which treatment would be most effective.

A benefit of the mechanistic modelling approach proposed here is that the underlying mechanisms of the model can be extended and modified as new scientific knowledge and data are generated. Since the parameters of the model have mechanistic interpretations, the model can be used to investigate the effects of different biological parameters. Changes can also be made to model different situations, such as incorporating additional mechanisms to represent treatments. Future work may use pharmacokinetic (PK) data to describe how the within-host concentration of the treatment changes in time, and pharmacodynamic (PD) data to define an extracellular killing rate of bacteria as a function of antibiotic concentration [44]. It would then be possible to use the resulting model to quantify the effect that treatment decisions (such as the initial timing, frequency, or dose) have on reducing the probability of infection.

## Methods

### Mathematical model

The within-host model of inhalational anthrax shown in Fig 1 is a hybrid model comprising both discrete and continuous variables. A multi-dimensional continuous-time Markov chain (CTMC) is used to describe the discrete variables representing the numbers of extracellular spores, infected cells, and bacterial colony-forming units (CFU) within various compartments of the body following exposure. On the other hand, ordinary differential equations (ODEs) describe the continuous variables representing the amounts of the protective antigen (PA) component of the anthrax toxin. The compartments considered are the airways of the lungs (A), the draining lymph nodes of the lungs (LN), and the blood/circulation (C). The model contains the following variables:

- $S_A(t)$ is the number of extracellular spores in the airways of the lungs.
- $P(t)$ is the number of infected phagocytes migrating from the lungs to the draining lymph nodes.
- $B_{LN}(t)$ is the amount of extracellular vegetative bacteria in the lymph nodes compartment, measured in CFU.
- $B_C(t)$ is the amount of extracellular vegetative bacteria in the blood compartment, measured in CFU.
- $PA_{LN}(t)$ is the amount of PA (units ng) in the lymph nodes compartment.
- $PA_C(t)$ is the amount of PA in the blood compartment.

The model is used to simulate an individual's infection time course following inhalation of a number of *B. anthracis* spores. After inhalation, spores are transported through the respiratory system and can become deposited in different areas, or may be exhaled without becoming deposited. Hence the initial dose of spores that is delivered to the lungs is generally smaller than the original inhaled dose [45]. This is captured in the model by specifying that each inhaled spore becomes deposited in the lungs with probability $\hat{\phi}$. In the model, time $t = 0$ represents the time at which inhaled spores become deposited in the lungs. The initial number of spores deposited in the airways is sampled from a binomial distribution, $S_A(0) \sim B(D, \hat{\phi})$, where $D$ is the dose of spores that the individual is exposed to through inhalation, and $\hat{\phi}$ is the probability that a given inhaled spore is deposited in the lungs [46]. These spores are then removed from the lung airways at rate $\rho$. For each spore that exits the airways, it is either cleared from the host (and therefore has no further chance to cause infection) with probability $1-q$, or it successfully infects a phagocyte with probability $q$.

The variable $P$ represents the number of phagocytes (each assumed to be initially infected with a single spore) that will eventually rupture in the lymph nodes and release some bacteria. The time taken for a given infected phagocyte to migrate to the lymph nodes and rupture is assumed to follow an Erlang distribution with shape parameter $n = 3$ and rate parameter $\delta$. This is implemented in the model by partitioning the infected phagocyte population into three stages such that $P(t) = \sum_{i=1}^{3} P_i(t)$, where $P_i(t)$ represents the number of infected phagocytes in stage $i \in \{1, 2, 3\}$. Each phagocyte begins in stage 1 and passes through the three stages in order, where the time spent in each stage is exponentially-distributed with rate $\delta$. It then releases some (random) amount of vegetative bacteria into the lymph nodes upon leaving the third stage. The random variable for the number of CFU released from one of these infected phagocytes, referred to here as the rupture size, is assumed to follow a geometric distribution. That is, the probability that the rupture size of a cell initially infected with one spore is equal to $k \in \mathbb{N}$ is given by $\frac{1}{R}\left(1 - \frac{1}{R}\right)^{k-1}$, where $R$ is the mean of this geometric rupture size distribution. This is based on a model at the intracellular scale, which was proposed and calibrated in [14].

In the lymph nodes compartment, extracellular bacteria replicate at rate $\lambda_{LN}$, and die at rate $\mu_{LN}$. A second, density-dependent death rate is also included, which increases with the number of bacteria in the compartment. This results in logistic growth of the bacterial population, representing the limited resources for proliferation.

If an infection becomes established in the lymph nodes and the number of bacteria continues to grow, this will lead to damage to the lymph nodes. It will also eventually result in an inflammatory response, which will cause the tissues and blood vessels to become leaky in order to facilitate an influx of immune system cells. However, this can also allow bacteria to escape from the lymph nodes and enter the circulation. This is represented in the model by assuming that once the amount of bacteria in the lymph nodes reaches a threshold, $M$ (representing a certain level of damage to the lymph nodes), bacteria begin to migrate to the blood

compartment at rate $m_{LN}$. In the circulation compartment, extracellular bacteria proliferate with net growth rate $\lambda_C$. This can be interpreted as the difference between the bacterial replication and death rates, but these are combined into a single parameter since the values of these individual rates would not be identifiable for this model. Although bacteria move through the blood to reach various organs, they are unlikely to spend much time in the blood until the disease is in the final bacteraemic stages. Therefore, the model assumes that bacteria are removed from the blood into tissues and organs at rate $m_C$.

The possible transitions considered in the CTMC part of the within-host model are shown in Table 3. Additionally, the amounts of PA (units ng) in the lymph nodes and blood compartments are given by the continuous variables $PA_{LN}(t)$ and $PA_C(t)$, respectively. These follow the ODEs

$$
\begin{aligned}
\frac{dPA_{LN}}{dt} &= \beta B_{LN} - (\mu_T + m_{LN})PA_{LN}, \\
\frac{dPA_C}{dt} &= m_{LN}PA_{LN} + \beta B_C - (\mu_T + m_C)PA_C.
\end{aligned}
\tag{4}
$$

The amount of PA increases from production at a rate proportional to the amount of extracellular bacteria, where $\beta$ is the production rate of PA per CFU. On the other hand, PA is removed due to natural degradation. It also interacts with host cell membranes and is cleaved by the host furin, to create a binding site for the active toxin components, LF and EF. To model these two removal mechanisms, the amount of PA decreases at a rate proportional to the current amount of PA, where $\mu_T$ is the rate of decrease from both natural decay and binding. The rates at which PA is transported between the different model compartments are assumed to be the same as those for the bacteria.

When making use of the model to compare to data, the stochastic model was simulated using the Gillespie algorithm [47] with the transition probabilities given in Table 3. The continuous variables, $PA_{LN}$ and $PA_C$, do not follow discrete transitions in the same way as the variables in Table 3. Instead, at each time step of the Gillespie algorithm, these continuous variables are updated by solving Eq (4) for the given time step, with initial conditions and the values of $B_{LN}$ and $B_C$ set to their values at the previous time step. Note that none of the rates of the discrete transitions in Table 3 depend on the variables $PA_{LN}$ and $PA_C$, which allows us to take this approach. The model is simulated in this way until either $B_C(t) \geq 1$ or $S_A(t) + P(t) + B_{LN}(t) = 0$. If $B_C(t) \geq 1$ is reached (which can only happen once the bacterial load in the lymph nodes is above the migration threshold, $B_{LN}(t) \geq M$), the rest of the time course is simulated using an ODE system representing the mean-field model. The final state of the model from the stochastic simulation is taken as the initial condition for the mean-field ODE version of the model, which is given by the following differential equations:

$$
\begin{aligned}
\frac{dS_A}{dt} &= -\rho S_A, \\
\frac{dP_1}{dt} &= q\rho S_A - \delta P_1, \\
\frac{dP_2}{dt} &= \delta P_1 - \delta P_2, \\
\frac{dP_3}{dt} &= \delta P_2 - \delta P_3, \\
\frac{dB_{LN}}{dt} &= \delta R P_3 + \lambda_{LN} B_{LN}\left(1 - \frac{B_{LN}}{K_{LN}}\right) - (\mu_{LN} + m_{LN}\mathbf{1}_{B_{LN}>M})B_{LN}, \\
\frac{dB_C}{dt} &= m_{LN}B_{LN}\mathbf{1}_{B_{LN}>M} + \lambda_C B_C\left(1 - \frac{B_C}{K_C}\right) - m_C B_C,
\end{aligned}
$$

**Table 3**. **The transitions and their corresponding rates in the Markov chain part of the within-host model in Fig 1.** The symbol $\mathbf{1}_A$ represents an indicator function, which is equal to 1 if $A$ is true, and 0 otherwise. Variables not directly involved in the corresponding reaction are not shown in the transition, since they remain constant during the event. The continuous variables for the amounts of PA (units ng) in the lymph nodes and blood compartments do not undergo discrete transitions in the same way as the other variables listed here but instead follow the ODEs in Eq (4).

| Transition | Rate |
|---|---|
| $S_A \longrightarrow S_A - 1$ | $(1-q)\rho S_A$ |
| $(S_A, P_1) \longrightarrow (S_A - 1,\ P_1 + 1)$ | $q\rho S_A$ |
| $(P_i, P_{i+1}) \longrightarrow (P_i - 1,\ P_{i+1} + 1)$ | $\delta P_i,\ i = 1, 2$ |
| $(P_3, B_{LN}) \longrightarrow (P_3 - 1,\ B_{LN} + k)$ | $\delta P_3 \frac{1}{R} \left(1 - \frac{1}{R}\right)^{k-1},\ k \in \mathbb{N}$ |
| $B_{LN} \longrightarrow B_{LN} + 1$ | $\lambda_{LN} B_{LN}$ |
| $B_{LN} \longrightarrow B_{LN} - 1$ | $\mu_{LN} B_{LN} + \lambda_{LN} \frac{B_{LN}^2}{K_{LN}}$ |
| $(B_{LN}, B_C) \longrightarrow (B_{LN} - 1,\ B_C + 1)$ | $m_{LN} B_{LN} \mathbf{1}_{B_{LN} > M}$ |
| $B_C \longrightarrow B_C + 1$ | $\lambda_C B_C$ |
| $B_C \longrightarrow B_C - 1$ | $\lambda_C \frac{B_C^2}{K_C} + m_C B_C$ |

$$\frac{dPA_{LN}}{dt} = \beta B_{LN} - (\mu_T + m_{LN}) PA_{LN},$$

$$\frac{dPA_C}{dt} = m_{LN} PA_{LN} + \beta B_C - (\mu_T + m_C) PA_C.$$

## Dose-response relationship

In order to use the model in Fig 1 to calculate the infection risk from exposure to different doses of spores, one can consider a simple discrete-time branching process approximation of the model. The timings of the events are ignored since we only need to focus on the chain of events that occurs, leading to a discrete-time process that represents the probabilities of different outcomes at each initial stage of the infection. We assume that the probability of response (*i.e.*, an established infection) for a given dose depends on the early-time dynamics in the airways and lymph nodes, but not in the blood.

There are two possible fates of each inhaled spore in the early stages of infection in the lungs and lymph nodes. These are established infection or clearance. Starting with a single inhaled spore, an infection will only become established if all three of the following occur. First, the inhaled spore must be deposited in the alveoli of the lungs, which occurs with probability $\hat{\phi}$. Then, this spore must be phagocytosed by a host cell that will ultimately release some positive number of bacteria into the lymph nodes, which occurs with probability $q$. If this happens, the infected host phagocyte migrates to the lymph nodes and releases $k \in \mathbb{N}$ bacterial CFU with probability $\frac{1}{R} \left(1 - \frac{1}{R}\right)^{k-1}$, where $R$ is the mean of this geometric rupture size distribution. Finally, this population of bacteria in the lymph nodes, starting at size $k$, must not become extinct.

In the early stages of infection, when the amount of bacteria in the lymph nodes is relatively small, one can neglect the density dependent term in the death rate of bacteria, so that bacteria are assumed to be independent of each other, with replication and death rates of $\lambda_{LN} > 0$ and $\mu_{LN} > 0$, respectively. Due to the presence of the migration threshold, $M$, it can also be assumed that migration of bacteria to the blood will not occur until the infection is established in the lymph nodes; therefore, migration is neglected in the very early stages. Thus, once released into the lymph nodes, extracellular bacteria can be killed by host

immune cells with probability $p = \frac{\mu_{LN}}{\mu_{LN} + \lambda_{LN}}$, or proliferate extracellularly with probability $1-p$. It is assumed that $p < 0.5$, otherwise the extracellular bacterial population would be certain to go extinct. The size of the bacterial population in the lymph nodes follows a simple one-dimensional random walk, with zero the only absorbing state, and probabilities $p$ and $1-p$ of moving from state $j$ to $j-1$ and $j+1$, respectively. Mathematically, we say that an infection is established if the bacterial population described by this discrete-time process approaches infinity rather than hitting zero. If the population starts in state $k$ after a rupture event, it is well known that the probability of ultimate extinction of this process is $\left(\frac{p}{1-p}\right)^k$. Therefore the probability that infection is established by a single inhaled spore is given by

$$
\begin{aligned}
r &= \hat{\phi}q \sum_{k=1}^{\infty} \frac{1}{R}\left(1 - \frac{1}{R}\right)^{k-1}\left(1 - \left(\frac{p}{1-p}\right)^k\right) \\
&= \hat{\phi}q \frac{1}{R}\left[\sum_{k=0}^{\infty}\left(1 - \frac{1}{R}\right)^k - \frac{p}{1-p}\sum_{k=0}^{\infty}\left(\frac{p}{1-p}\left(1 - \frac{1}{R}\right)\right)^k\right] \\
&= \hat{\phi}q \frac{1}{R}\left[R - \frac{Rp}{R(1-2p)+p}\right] \\
&= \frac{\hat{\phi}qR(1-2p)}{R(1-2p)+p}.
\end{aligned}
\tag{5}
$$

If multiple spores are initially inhaled, the probability of an infection becoming established is equal to the probability that not all of the inhaled spores are cleared (*i.e.*, at least one of the inhaled spores leads to an established infection). Since the initial spores are assumed to be independent of each other, the dose-response curve can therefore be described using the exponential dose-response model. That is, for a Poisson-distributed dose with mean $D$, the probability of infection is given by

$$
I(D) = 1 - e^{-rD},
\tag{6}
$$

where $r$ is the probability that a single spore will produce a response, given by Eq (5).

## Rabbit and guinea-pig data sets

We used data from a study by Gutting et al. [22] in which rabbits were infected with the highly virulent Ames strain of *B. anthracis* in order to characterise bacterial dissemination following exposure to a high aerosol dose of spores. The mean inhaled dose given was $4.428 \times 10^7$ spores. The data set contains the number of bacterial CFU in the tracheo-bronchial lymph nodes (TBLN) and blood of rabbits at times 1, 6, 12, 24, and 36 hours after exposure. Between 4 and 10 rabbits were sacrificed to obtain the measurements at each time point. For the measurements of CFU in the TBLN, Gutting et al. pooled all lymph nodes that they could find from the tracheal and bronchial regions, to obtain the number of CFU in the TBLN of each animal. However, the lymph nodes varied in size and number from animal to animal. For the blood measurements, Gutting et al. used a conversion factor of 56 millilitres of blood per kilogram of body weight to convert measurements of CFU/ml to total CFU counts in the circulation. The data used is provided in Table 4.

We also used data from a study by Savransky et al. [23] in which guinea pigs were infected with the Ames strain of *B. anthracis* using nose-only aerosol exposure. The dose given was specified by the authors as $2 \times 10^7$ spores. The data set contains measurements of CFU and PA in the blood of guinea pigs at times 24, 30, 36, 48, and 72 hours after exposure. Four animals

**Table 4. Data for the number of CFU present in the tracheobronchial lymph nodes (TBLN) and blood of rabbits at different times after exposure to a mean inhaled dose of** $4.428 \times 10^7$ **spores.** The number of CFU in the TBLN could not be quantified for all rabbits at 24h and 36h, due to the CFU being too numerous to count after plating. This is why we only have a lower bound for the number of CFU for some of the rabbits at these times. For the rabbits that died before the time point at which they were scheduled to be sacrificed, we assume that if they had survived then their CFU measurement would have been greater than the highest measurement observed for a surviving animal, which was $8.9 \times 10^7$ CFU in the TBLN and $8.4 \times 10^7$ CFU in the blood. These data were obtained during the study published by Gutting et al. [22]

| Time (hours) | CFU in TBLN | CFU in blood |
|---|---|---|
| 1 | 0 | 0 |
| | 0 | 0 |
| | 0 | 0 |
| | 0 | 0 |
| 6 | 0 | 0 |
| | 0 | 0 |
| | 0 | 0 |
| | 0 | 0 |
| | 0 | 0 |
| 12 | 0 | 0 |
| | 0 | 0 |
| | 17 | 0 |
| | 17 | 0 |
| | 18 | 0 |
| 24 | 0 | 0 |
| | 0 | 0 |
| | 0 | 0 |
| | 0 | 0 |
| | 0 | 0 |
| | $1.7 \times 10^3$ | $1.4 \times 10^6$ |
| | $1.4 \times 10^4$ | $1.3 \times 10^6$ |
| | $> 2.6 \times 10^4$ | $7.6 \times 10^5$ |
| | $> 2.7 \times 10^4$ | $2.0 \times 10^5$ |
| | $> 2.7 \times 10^4$ | $7.1 \times 10^5$ |
| 36 | 0 | 0 |
| | $3.0 \times 10^6$ | $2.6 \times 10^4$ |
| | $1.6 \times 10^7$ | $6.2 \times 10^4$ |
| | $4.4 \times 10^7$ | $8.4 \times 10^7$ |
| | $8.9 \times 10^7$ | $9.3 \times 10^6$ |
| | $> 2.6 \times 10^4$ | $3.0 \times 10^5$ |
| | $> 2.6 \times 10^6$ | $6.0 \times 10^5$ |
| | $> 2.7 \times 10^6$ | $1.2 \times 10^5$ |
| | died (assume $> 8.9 \times 10^7$) | died (assume $> 8.4 \times 10^7$) |
| | died (assume $> 8.9 \times 10^7$) | died (assume $> 8.4 \times 10^7$) |

were scheduled to be sacrificed at each time point. Savransky et al. measured the number of CFU per ml of blood, for which the lower limit of detection was 100 CFU/ml. They also measured the amount of PA in sera. The lower limit of detection for the PA measurements was 1.3 ng/ml. The average circulating blood volume of a guinea pig is reported to be 75ml per kg of body weight [48], and the guinea pigs used in the study by Savransky et al. weighed between 635g and 874g. Therefore, to convert the concentrations of CFU and PA to the total amounts in the circulation, we assumed a value of 57ml $\approx$ 75ml/kg $\times$ 0.7545kg for the circulating blood volume of a typical guinea pig used in the study. The data used is reported in Table 5.

**Table 5. Data for the number of CFU and amount of PA present in the blood of guinea pigs at different times after exposure to a mean dose of** $2 \times 10^7$ **spores.** For the guinea pigs that died before the time point at which they were scheduled to be sacrificed, we assume that if they had survived then their CFU and PA measurements would have been greater than the highest measurements observed for a surviving animal, which were $2.5 \times 10^{10}$ CFU and $1.5 \times 10^6$ ng of PA. These data have been obtained from [23, Fig 3] and converted to the amounts in the whole blood compartment by assuming a volume of 57 ml of blood in a guinea pig.

| Time (hours) | CFU in blood | PA in blood (ng) |
|---|---|---|
| 24 | $< 5.7 \times 10^3$ | $<74.1$ |
|  | $< 5.7 \times 10^3$ | $<74.1$ |
|  | $< 5.7 \times 10^3$ | $<74.1$ |
|  | $< 5.7 \times 10^3$ | $<74.1$ |
| 30 | $< 5.7 \times 10^3$ | $<74.1$ |
|  | $5.7 \times 10^3$ | $<74.1$ |
|  | $7.1 \times 10^3$ | $<74.1$ |
|  | $1.1 \times 10^4$ | 234.8 |
| 36 | $< 5.7 \times 10^3$ | $<74.1$ |
|  | $2.1 \times 10^4$ | 83.1 |
|  | $2.2 \times 10^4$ | 405.8 |
|  | $4.0 \times 10^4$ | 548.9 |
| 48 | $1.1 \times 10^4$ | $<74.1$ |
|  | $3.4 \times 10^5$ | $2.8 \times 10^3$ |
|  | $7.4 \times 10^6$ |  |
|  | died (assume $> 2.5 \times 10^{10}$) | died (assume $> 1.5 \times 10^6$) |
| 72 | $2.5 \times 10^{10}$ | $1.5 \times 10^6$ |
|  | died (assume $> 2.5 \times 10^{10}$) | died (assume $> 1.5 \times 10^6$) |
|  | died (assume $> 2.5 \times 10^{10}$) | died (assume $> 1.5 \times 10^6$) |
|  | died (assume $> 2.5 \times 10^{10}$) | died (assume $> 1.5 \times 10^6$) |

## Rabbit model calibration

Prior to the parameter inference, we fixed the values of some model parameters that could be informed from the literature. For example, using a combination of particle deposition data and Regional Deposited Dose Ratio (RDDR) software, Gutting [17] estimated the deposition probability for inhaled *B. anthracis* spores in the tracheo-bronchial and pulmonary regions of rabbit lungs to be $\hat{\phi} = 0.092$. We have therefore fixed this value of the deposition probability in our rabbit model. Furthermore, Gutting [17] used data to estimate values for the rates of two removal mechanisms of spores from rabbit airways; namely, the mucociliated (physical) clearance rate, $k_c = 0.0628$ h$^{-1}$, and the rate of transport of spores from the lung airways into the lung tissue, $k_p = 0.0107$ h$^{-1}$. We have made use of these estimates to set a value for the overall rate of removal of spores from the airways in our model, given by $\rho = k_c + k_p = 0.0735$ h$^{-1}$.

We used the (conditional) rupture size distribution given by the best-fitting parameters from the intracellular model calibration in [14]. This is a geometric distribution with a mean of $R = 1.6$ CFU (*i.e.*, if a phagocyte engulfs a spore and subsequently ruptures, it is assumed to release 1.6 CFU on average). This is very similar to the (per spore) rupture size in the model by Day et al. [16], where it was assumed that a phagocyte would release 5 CFU into the lymph nodes after engulfing 3 spores in the lungs.

We set the growth rate of bacteria in the circulation compartment of rabbits to be $\lambda_C = 0.17$ h$^{-1}$. This was estimated from data of *B. anthracis* growth in rabbit sera from [49]. We also fixed $K_{LN} = 10^9$ CFU and $K_C = 10^{11}$ CFU. It is necessary to include these carrying capacities in order to ensure that the model does not predict uncontrolled growth of extracellular bacteria. However, saturation of the bacterial populations is not observed in the data during the time period of observation, so it would not be possible to identify a value for these parameters with the available data.

The model in Fig 1 was simultaneously fitted to rabbit dose-response data and *in vivo* numbers of bacteria, using Approximate Bayesian Computation Sequential Monte Carlo (ABC-SMC) [24], in order to estimate the posterior distribution of the parameter vector ($q$, $\delta$, $\lambda_{LN}$, $\mu_{LN}$, $M$, $m_{LN}$, $m_C$). Values of these parameters were sampled from log-uniform prior distributions.

For each sampled parameter set, the exponential dose-response curve was calculated according to Eqs (5) and (6), to obtain the probabilities of infection $p_D = I(D)$ for each dose $D$ in the dose-response data set. Then for each dose $D$, a number of infections was sampled from a binomial distribution, $B(N_D, p_D)$, where $N_D$ is the size of the exposed group for dose $D$ in the data set. The endpoint of the dose-response study was lethality, whereas we refer to $p_D$ as the probability of infection (by which we mean development of pathological disease). However, for rabbits, established inhalational anthrax will very often lead to death in the absence of treatment, so we make the simplifying assumption that the probability of death is equivalent to the probability of infection [25]. Therefore, the Euclidean distance was computed between the set of sampled numbers of infections from the model and the observed numbers of deaths in each group from the data, giving a distance measure between the model and the dose-response data for a particular parameter set.

To obtain a distance measure between the model and the *in vivo* bacterial counts from Table 4, a different stochastic realisation of the model was compared to the observation of the number of CFU in the TBLN and blood for each animal, using the Euclidean distance between the logarithms of the model output and observed data. For time points at which there were animals with no CFU observed in the lymph nodes, the smallest simulated values from the set of model realisations were compared with these unobserved data points. In the cases of animals where we only have a lower bound on the observed number of CFU, the distance between the model output and this observation was taken to be zero if the model output was above this lower bound. Otherwise, the Euclidean distance between the logarithms of the model output and the lower bound was taken.

## Guinea-pig model calibration

We simultaneously fitted the model in Fig 1 to the guinea-pig dose-response data from Gutting et al. [21] and the *in vivo* measurements of bacterial CFU and PA from Savransky et al. [23], provided in Table 5.

For each sampled parameter set, we compared the exponential dose-response curve (computed using Eqs (5) and (6)) to the guinea-pig dose-response data in the same way as described above for the rabbit model calibration. We also compared four stochastic model realisations to the four data observations at each time point in Table 5, where the model outputs were ordered at each time point so that the smallest model outputs were compared with the smallest data points. For measurements that were below the lower limit of detection, the distance between the model output and the observation was taken to be zero if the model output was below this limit of detection. Otherwise, the Euclidean distance between the logarithms of the model output and the limit of detection was taken.

## Simulating human infection from exposure to symptom onset

To simulate human infection, we use the simplified model in Fig 7, which considers only the lung airways and a single 'whole body' compartment in which extracellular bacteria arrive due to rupture events, replicate at rate $\lambda$, and die at rate $\mu$. The total amount of extracellular bacteria at time $t$ post exposure is denoted by $B(t)$ (see Eq (7)). Here we describe the method that we use to simulate the incubation period (time between exposure and symptom onset) for an

individual from the Sverdlovsk outbreak data set, in order to compare our model output with the data.

In order to simulate the infection of an individual, we first need to sample the dose of spores, $D$, that the individual is initially exposed to. Wilkening [31] used an atmospheric dispersion model to simulate the Sverdlovsk release and estimated the dose to which the victims would have been exposed. We have assumed that the distribution of exposed doses among the victims follows the empirical distribution obtained by Wilkening for their 'Model D' (see [31, Fig 11]), since our model corresponds to the same exponential dose-response curve. Given an exposure dose, $D$, sampled from this distribution, the initial number of spores that are deposited in the lungs and will eventually be phagocytosed and lead to a rupture event, $S_P(0)$, is sampled from the binomial distribution $B(D, \phi)$.

Since we want to use the model to simulate the incubation period of an infected individual, we also need to specify how 'symptom onset' is represented in the model. For example, one could specify a threshold for the number of bacteria or the toxin concentration at which symptoms are assumed to first appear. For this illustration of the model, we assume that symptom onset occurs when the bacterial load is $10^{10}$ CFU, which is the same threshold value used by Wilkening [32].

After inhalation of *B. anthracis* spores, it is possible that the infection will be cleared before symptom onset occurs. However, as the bacterial load increases, clearance of the bacteria becomes less likely. In particular, the probability of clearance starting from a bacterial load of $B(t) = k$ is $\left(\frac{\mu}{\lambda}\right)^k$ (given that $\mu < \lambda$). We make the assumption that once the bacterial load reaches a level of $b$ CFU such that $\left(\frac{\mu}{\lambda}\right)^b < 10^{-5}$, the individual will not recover. Therefore the model is simulated as a Markov chain until either $B(t) \geq b$ or all populations (spores in the airways, infected phagocytes, extracellular bacteria) become extinct. If the threshold $b$ is reached, the current state of the model is then taken as the initial condition for the following mean-field ODE version of the model:

$$\frac{dS_P}{dt} = -\rho S_P,$$
$$\frac{dP_1}{dt} = \rho S_P - \delta P_1,$$
$$\frac{dP_2}{dt} = \delta P_1 - \delta P_2,$$
$$\frac{dP_3}{dt} = \delta P_2 - \delta P_3,$$
$$\frac{dB}{dt} = \delta R P_3 + (\lambda - \mu)B. \tag{7}$$

Note that here we are only considering the spores that will eventually cause a rupture event in the lymph nodes. That is, the variable $S_P$ represents the number of spores in the lung airways that will eventually be phagocytosed and cause the infected phagocyte to rupture in the lymph nodes.

If the infection is cleared before the bacterial load reaches $b$ CFU, then the incubation period is considered to be 'infinite'. Otherwise, the incubation period is recorded as the time that the bacterial load reaches the symptom onset threshold (*i.e.*, $B(t) = 10^{10}$ CFU).

## Acknowledgments

We would like to thank Dr. Bradford Gutting for providing us with the raw data from the study in [22] and for very useful discussions. This manuscript has been internally reviewed

at Los Alamos National Laboratory, and assigned the reference number LA-UR-25-23417 (CMP).

## Author contributions

**Conceptualization:** Bevelynn Whaler, Grant Lythe, Joseph J. Gillard, Thomas R. Laws, Jonathan Carruthers, Thomas Finnie, Carmen Molina-París, Martín López-García.

**Formal analysis:** Bevelynn Whaler, Martín López-García.

**Funding acquisition:** Grant Lythe, Joseph J. Gillard, Thomas R. Laws, Carmen Molina-París, Martín López-García.

**Investigation:** Bevelynn Whaler, Grant Lythe, Joseph J. Gillard, Thomas R. Laws, Jonathan Carruthers, Thomas Finnie, Carmen Molina-París, Martín López-García.

**Methodology:** Bevelynn Whaler, Martín López-García.

**Project administration:** Martín López-García.

**Supervision:** Grant Lythe, Joseph J. Gillard, Thomas R. Laws, Carmen Molina-París, Martín López-García.

**Validation:** Bevelynn Whaler, Martín López-García.

**Visualization:** Bevelynn Whaler, Martín López-García.

**Writing – original draft:** Bevelynn Whaler.

**Writing – review & editing:** Bevelynn Whaler, Grant Lythe, Joseph J. Gillard, Thomas R. Laws, Jonathan Carruthers, Thomas Finnie, Carmen Molina-París, Martín López-García.

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
