## [Decision Letter · Decision Letter 0]

13 Jul 2025

 PCOMPBIOL-D-25-01117

Mechanistic within-host mathematical model of inhalational anthrax

PLOS Computational Biology

Dear Dr. Whaler,

Thank you for submitting your manuscript to PLOS Computational Biology. The manuscript as is was embraced enthusiastically. We would only ask that the authors please address Reviewer 2's concern regarding the controversial incubation times and the minor revisions noted by all three reviewers. We are, therefore, excited to invite you to submit a revised version of the manuscript that addresses these points.

Please submit your revised manuscript within 60 days Sep 12 2025 11:59PM. If you will need more time than this to complete your revisions, please reply to this message or contact the journal office at ploscompbiol@plos.org. Please include the following items when submitting your revised manuscript:

We look forward to receiving your revised manuscript.

Kind regards,

Brittany Rife Magalis, Ph.D

Academic Editor

PLOS Computational Biology

James Faeder

Section Editor

PLOS Computational Biology

**Additional Editor Comments:**

The submitted manuscript was embraced enthusiastically. I would only as that the authors please address Reviewer 2's concern regarding the controversial incubation times and the minor revisions noted by all three reviewers.

**Journal Requirements:**

3) Please amend your detailed Financial Disclosure statement. This is published with the article. It must therefore be completed in full sentences and contain the exact wording you wish to be published.

1) State the initials, alongside each funding source, of each author to receive each grant. For example: "This work was supported by the National Institutes of Health (####### to AM; ###### to CJ) and the National Science Foundation (###### to AM).".

4) Kindly revise your competing statement to align with the journal's style guidelines: 'The authors declare that there are no competing interests.'

**Reviewers' comments:**

Reviewer's Responses to Questions

**Comments to the Authors:**

Reviewer #1: This is an interesting paper that considers an important problem of efficiently modelling within-host dynamics of bacterial anthrax. Using a clear mechanistic structure for disease transmission/progression in different parts of the body, they are then able to not just qualitative reproduce available data, but to actually fir their results to a number of model organisms, including rabbit, guinea pig, and even a known human outbreak. Overall, the paper is clear and well-written, and I am happy to recommend publication, provided the authors address the following minor issues related to improving clarity/readability.

1. While units of measurement for different parameters are stated in Table 1, perhaps it would be a good idea to also include them next to parameters themselves in the leftmost column of Table 2.

2. When introducing Sverdlovsk data on p. 13, it is advisable to mention straightaway the limitations of those data in terms of possible use of tetracycline in some patients, but with an unknown start date and unknown duration.

3. Perhaps, it would be a good idea to mention in Table 3 transitions for PA_{LN} and PA_C in the same way, as they are described for all other model compartments. Also, to be absolutely sure, do I understand correctly that the ODE system representing the mean-field model was not actually used for simulations, and instead it was rather the stochastic version that was simulated using Gillespie algorithm with transition probability as given in Table 3?

Reviewer #2: This is an interesting study on applying a mechanistic mathematical model of anthrax infection to data from experimental infections of animals. The insights from the animal data-based model are also applied toward exploring a model of human incubation time estimates, which is also interesting and useful. The details for the within-host model are novel extensions to prior models in this field, leading to novel conclusions.

While I have not checked all the details of the analysis and results, I have no reason to doubt that they were done correctly. The model and analysis techniques are well explained, the codes to reproduce the analysis are publicly available and appear to be well documented, and the results are plausible and not inconsistent with prior, related work. I think the model and techniques described are quite useful advancements for this area of work.

My only suggestion for improvement is regarding the human incubation time data from the 1979 Sverdlovsk infections. The authors use data from a 2006 manuscript by Wilkening, but there was later work that arguably produced a more rigorously derived list of incubation times. A 2012 study by National Academies Institute of Medicine (https://nap.nationalacademies.org/catalog/13218/prepositioning-antibiotics-for-anthrax) cast doubt on some of the incubation times from the Wilkening data set, particularly a few very short (< 4 days) incubation times that were from individuals with unconfirmed Anthrax infection. They judged that a list of incubation times from 31 autopsy-confirmed anthrax victims reported by Abramova et al. (https://doi.org/10.1073/pnas.90.6.2291) provided a stronger evidence base for incubation period distribution from this event.

I wonder if using applying the authors’ analysis to this set of 31 incubation times would lead to different conclusions compared to the result from the Wilkening data. I suggest the authors attempt this comparison if possible.

Reviewer #3: This was an excellent manuscript to read - very clearly written, solid modelling assumptions, with open and honest interpretation of the work, as well useful suggestions for future work. The authors were able to calibrate their model against experimental data, whilst reflecting on the limitations of their data and approach. Useful insights were generated, but again placed in an honest framework of what more needs to be done before the findings can be used in practice.

I have only two very minor points:

- the assumption that only 1CFU is produced by one chain of bacteria seems odd since the authors describe the chains of bacteria as being very long. Why would only 1CFU emerge from this? There's some discussion of this in the Methods, but a little more discussion here would be interesting.

- in Figure 2, two of the data points lie obviously outside of the range of the simulations. Can the authors comment on these? It's not unusual for the simulations to not be able to capture all the data (as is also the case elsewhere in the manuscript), but these two appear more extreme than data points later on that aren't captured.

**Have the authors made all data and (if applicable) computational code underlying the findings in their manuscript fully available?**

Reviewer #1: Yes

Reviewer #2: Yes

Reviewer #3: Yes

PLOS authors have the option to publish the peer review history of their article (what does this mean?). If published, this will include your full peer review and any attached files.

Reviewer #1: **Yes: **Prof Konstantin Blyuss

Reviewer #2: No

Reviewer #3: No

**Figure resubmission:**
---

## [Editor Report · Decision Letter 1]

14 Aug 2025

Dear Dr Whaler,

We are pleased to inform you that your manuscript 'Mechanistic within-host mathematical model of inhalational anthrax' has been provisionally accepted for publication in PLOS Computational Biology.

Best regards,

Brittany Rife Magalis, Ph.D

Academic Editor

PLOS Computational Biology

James Faeder

Section Editor

PLOS Computational Biology

---

## [Editor Report · Acceptance letter]

PCOMPBIOL-D-25-01117R1

Mechanistic within-host mathematical model of inhalational anthrax

Dear Dr Whaler,

I am pleased to inform you that your manuscript has been formally accepted for publication in PLOS Computational Biology. Your manuscript is now with our production department and you will be notified of the publication date in due course.

With kind regards,

Benedek Toth
